# Locality in Image Diffusion Models Emerges from Data Statistics

**Artem Lukoianov**
Massachusetts Institute of Technology
arteml@mit.edu

**Chenyang Yuan**
Toyota Research Institute
chenyang.yuan@tri.global

**Justin Solomon**
Massachusetts Institute of Technology
jsolomon@mit.edu

**Vincent Sitzmann**
Massachusetts Institute of Technology
sitzmann@mit.edu

https://locality.lukoianov.com

## Abstract

Recent work has shown that the generalization ability of image diffusion models arises from the locality properties of the trained neural network. In particular, when denoising a particular pixel, the model relies on a limited neighborhood of the input image around that pixel, which, according to the previous work, is tightly related to the ability of these models to produce novel images. Since locality is central to generalization, it is crucial to understand why diffusion models learn local behavior in the first place, as well as the factors that govern the properties of locality patterns. In this work, we present evidence that the locality in deep diffusion models emerges as a statistical property of the image dataset and is not due to the inductive bias of convolutional neural networks, as suggested in previous work. Specifically, we demonstrate that an optimal parametric linear denoiser exhibits similar locality properties to deep neural denoisers. We show, both theoretically and experimentally, that this locality arises directly from pixel correlations present in the image datasets. Moreover, locality patterns are drastically different on specialized datasets, approximating principal components of the data's covariance. We use these insights to craft an analytical denoiser that better matches scores predicted by a deep diffusion model than prior expert-crafted alternatives. Our key takeaway is that while neural network architectures influence generation quality, their primary role is to capture locality patterns inherent in the data.

## 1 Introduction

Denoising diffusion models [8, 27] have achieved state-of-the-art results for generative modeling, especially for domains involving continuous data distributions such as images, videos, and audio. They are trained to predict a clean image from one corrupted by varying levels of Gaussian noise. Analysis of the diffusion model training objective leads to an apparent paradox: The objective admits a unique, analytical, non-parametric, and closed-form solution that is only a function of the training dataset. However, this so-called *optimal denoiser* does not empirically match the outputs of deep diffusion models and in fact can only produce images in the training set, exhibiting perfect "memorization" and failing to generate novel images.

Recent work investigates this paradox and proposes changes to the optimal denoiser to close the gap between theory and practice [12, 19, 25, 26]. Kamb and Ganguli [12] hypothesize that inductive biases of the neural network architecture—particularly shift equivariance and locality biases of

convolutional neural networks—prevent the model from converging to the global optimum of the loss function and thus allow for generalization. Locality here means that during denoising, any pixel in the denoised output will only be sensitive to a local neighborhood around that pixel in the noisy input. They demonstrate that adding locality and equivariance constraints to the closed-form optimal denoiser yields a model that generates images that closely resemble those generated by a simple U-Net diffusion model. However, their theory has a key limitation: it cannot predict the degree of locality from first principles. Instead, their method relies on *measuring* the receptive field of a trained U-Net diffusion model and estimating a patch size for each diffusion timestep that is then fed as a parameter to their analytical model. Further, other neural network architectures, such as transformers, can similarly generate novel images, but they lack either explicit locality or shift-equivariance inductive biases; even U-Nets generally have a receptive field that covers the complete image.

In this work, we demonstrate that locality in image diffusion models is *not* a property of the neural network architecture and instead can be derived directly from the training dataset via simple statistical analysis. Specifically, we analyze the principal components of the data—in particular, their signal-to-noise (SNR) ratio—and show that learned sensitivity fields for different architectures closely approximate projection operators onto principal components with high SNR. On CIFAR10 [16], a dataset with high self-similarity across pixel locations, a simple locality pattern emerges. However, on CelebA-HQ [13], a dataset of centered human faces, sensitivities are nonlocal, aligned with correlations of pixels across, for instance, the eyes. We relate this behavior to the optimal linear denoiser known as the Wiener filter, establishing a connection to prior work that observed linear behavior of diffusion models [17, 29]. We provide further evidence that sensitivity fields are learned to match statistical properties of the training set by showing that we can achieve arbitrary, nonlocal patterns in a model's sensitivity field by imperceptibly editing the pixel statistics of the training set.

We rigorously benchmark recent analytical models by how well they match generations by a trained deep diffusion model. Surprisingly, we find that a simple Wiener filter *outperforms* all recent analytical methods based on modifications of the optimal denoiser. Integrating our analytically-derived sensitivity fields into the model of Kamb and Ganguli [12], however, yields the best-performing analytical diffusion model to date across multiple datasets, including CIFAR10 [16], AFHQv2 [4], and CelebA-HQ [13]. This observation provides evidence that capturing pixel correlations across a dataset plays a major role in the performance of denoising diffusion models.

In summary, our contributions are as follows:

- We demonstrate that local sensitivities in trained image diffusion models are a learned property of deep diffusion models and *not* just an inductive bias of the model architecture.
- We analytically derive the spatial sensitivity of an optimal linear filter as a function of the training data and show empirically that it closely matches that learned by a denoising neural network, yielding both local and nonlocal sensitivities depending on the training data statistics.
- We establish a quantitative benchmark to measure how well an analytical diffusion model explains predictions made by a U-Net and demonstrate that, surprisingly, prior optimal denoiser-based methods are outperformed by a simple optimal linear filter.
- We incorporate our analytically-computed locality into the optimal denoiser-based model proposed by Kamb and Ganguli [12] and show that it outperforms alternative analytical models while eliminating a previously heuristically-determined hyperparameter and the need to measure it from pre-trained neural networks.

## 2   Preliminaries and Related Work

We discuss preliminaries and related work for our primary line of inquiry, building analytical models of deep diffusion networks.

**Denoising diffusion models.**   Score-based image generative models [8, 27, 28] learn to reverse the process of adding Gaussian noise to clean data. During training, we sample a data point $x_0$ from the training data distribution $X$, a noise level $t$ from the interval $[0, 1]$, and a Gaussian noise direction $\epsilon \sim N(0, I)$; a noise schedule $\alpha_t$ is chosen such that $\alpha_0 = 1$ and $\alpha_1 = 0$. We then add noise to $x_0$ to obtain $x_t = \sqrt{\alpha_t} x_0 + \sqrt{1 - \alpha_t} \epsilon$. The training objective for an image diffusion model $f(x, t)$, also

known as score-matching, aims to predict $x_0$ given $x_t$:[1]

$$\min_f \mathop{\mathbb{E}}_{\substack{x_0 \sim X \\ \epsilon \sim N(0,I) \\ t \sim [0,1]}} \left\| f\big(\sqrt{\alpha_t}\, x_0 + \sqrt{1-\alpha_t}\, \epsilon, t\big) - x_0 \right\|_2^2 \tag{1}$$

Recent studies explored the generalization capabilities of diffusion models, highlighting the contradiction between their theoretical propensity for memorization and their empirical ability to generate novel samples. Yoon et al. [34] introduce the *memorization-generalization dichotomy*, positing that diffusion models generalize when they avoid memorizing training data. Yi et al. [33] formalize generalization through mutual information metrics, demonstrating that trained diffusion models can generalize beyond the empirical optimal solutions. Gu et al. [7] further investigate this phenomenon, revealing that factors such as dataset size and conditioning can influence the extent of memorization in diffusion models.

**Analytical diffusion via the optimal denoiser.**   Multiple works [5, 14, 23, 25] identify the *optimal denoiser* as a promising analytical model for the behavior of deep diffusion models. This optimal denoiser $\hat{f}(x,t)$ minimizing (1) can be written as a conditional expectation:

$$\hat{f}(x,t) = \mathbb{E}[x_0 \mid x_t = x] \tag{2}$$

When the data distribution is approximated with a finite empirical distribution $X = \{x_0^{(i)}\}_{i \in [N]}$, and due to the fact that we have an analytic form for density $p(x_t) = N(\sqrt{\alpha_t}x_0, (1-\alpha_t)I)$, the optimal denoiser is available in closed-form [5, 14, 23, 25]:

$$\hat{f}(x,t) = \sum_{i=1}^N w_i(x,t) x_0^i, \quad w_i(x,t) = \mathrm{softmax}_i \left\{ -\frac{1}{2\sigma_t^2} \left\| \frac{1}{\sqrt{\alpha_t}} x - x_0^j \right\|^2 \right\}_{j \in [N]}, \tag{3}$$

where $\sigma_t^2 = (1-\alpha_t)/\alpha_t$ and $\mathrm{softmax}_i \{a_j\}_{j \in [N]} = \frac{\exp(a_i)}{\sum_{j=1}^N \exp(a_j)}$. This expresses the optimal denoiser as a kernel-weighted average over the training set, and clarifies the key limitation of the optimal denoiser as an appropriate model for deep neural networks: as the noise level approaches zero, the softmax term in effect picks the nearest neighbor in the training set. As a result, the optimal denoiser will always generate an image in the training set and never generate a novel image.

**Improving the optimal denoiser via smoothing.**   To promote generalization, Scarvelis et al. [25] propose smoothing the score function to generate novel samples different from the training data. Niedoba et al. [18] derive an efficient nearest neighbor search to support the implementation of the analytical score models. Separately, Shah et al.[26] propose smoothing the empirical training data distribution by adding Gaussian noise to encourage neural network models to generalize in the small-data regime. Aithal et al. [1] argue that hallucinations in diffusion models happen due to smooth interpolation between modes of the data distribution. Simply smoothing the score function, however, leads to pixel-space interpolation between training images and does not explain the high-quality, sharp novel images observed in practice.

**Adding inductive biases to the optimal denoiser.**   Kadkhodaie et al. [11] observe that the inductive biases of neural networks align well with the data density, effectively projecting data onto a low-dimensional basis adapted to the image structure. Based on this insight, Kamb and Ganguli [12] and Niedoba et al. [19] suggest that the gap between the optimal denoiser and deep diffusion models stems from the inductive bias of the deep neural network used to approximate the denoiser. Specifically, they assume that the structure of the convolutional U-Net [24] imposes constraints of locality and/or shift equivariance on the function $f$ in the score-matching objective eq. (1):

$$\min_f \mathop{\mathbb{E}}_{x_0 \sim X, \epsilon \sim N(0,I), t \sim [0,1]} \|f(\sqrt{\alpha_t}x_0 + \sqrt{1-\alpha_t}\epsilon, t) - x_0\|_2^2 \tag{4}$$

$$\text{s.t. } f^q(x,t) = f^q(M_t^q x, t) \,\forall q \in [Q] \qquad \text{(locality)}$$

$$f(g \circ x, t) = g \circ f(x,t) \,\forall g \in T(2), \qquad \text{(equivariance)}$$

---

[1]In practice, often a linear combination of $\epsilon$ and $x_0$ is predicted, but these are all equivalent to (1) for the purposes of deriving an optimal denoiser and for our theoretical analysis. Without loss of generality, through the rest of the paper, we will assume that all models are trained to predict $x_0$. We provide additional details on the effects of parametrization in Appendices A.1 and B.4

where $f^q$ is pixel $q$ (out of $Q$ total pixels) of the output, $M_t^q$ is a masking operator for each time-step $t$ selecting a patch around pixel $q$, and $T(2)$ is the 2D translation group acting on $x \in \mathbb{R}^n$ through $\circ$. They show that for general $M_t^q$, the solution for the constrained score-matching objective eq. (4) has a similar form compared to eq. (3), but with weights specific to each pixel $q$ such that

$$\hat{f}^q(x,t) = \sum_{\substack{i \in [N] \\ g \in T(2)}} w_{i,g}^q(x,t)(g \circ x_0^i)^q, \quad w_{i,g}^q(x,t) = \operatorname*{softmax}_{i,g} \left\{ -\frac{1}{2\sigma_t^2} \left\| M_t^q \left( \frac{1}{\sqrt{\alpha_t}} x - h \circ x_0^j \right) \right\|_2^2 \right\}_{\substack{j \in [N] \\ h \in T(2)}},$$
(5)

where $M_t^q$ is a binary mask, and we end up again with an isotropic multivariate Gaussian projected to a subspace defined by the mask. Effectively, this model splits each training image into patches of size $M_t$, forgets about their location (equivariance), and denoises each input pixel by taking the average of the center pixels in the ground truth patches weighted with the distances to them from the patch centered at $q$. Kamb and Ganguli [12] obtain patches for each noise level $t$ by iterating over all possible square binary patches and choosing the one that yields the best correlation with a trained diffusion model. Niedoba et al. [19] relax the constraint on the shape of the patches and allow them to be of arbitrary shape but compact and averaged across all pixels. Then they measure the average sensitivity of a trained U-Net and binarize it to get the masks. Both works fit masks $M_t^q$ to the empirically-observed locality fields of the trained models and average all the masks for each pixel $q$, i.e. $M_t^q$ and $M_t^p$ are identical up to a translation.

**Linear denoisers.** Recent studies have uncovered that diffusion models exhibit strong linear behavior. Wang and Vastola [29, 30] provide theoretical and empirical evidence that, at high noise levels, the learned score functions of well-trained diffusion models closely align with those of linear models. Linear denoisers are well-studied [2, 20, 31], and one can show that the optimal denoiser constrained to be linear has the same form as the optimal denoiser on a Gaussian dataset [15]. Li et al. [17] demonstrate that, particularly in the generalization regime and for high levels of noise diffusion, denoisers approximate the optimal denoiser for a multivariate Gaussian distribution characterized by the empirical mean and covariance of the dataset.

## 3 Deriving Denoising Sensitivity from Dataset Statistics

In this section, we explore the relationship between generalization and locality in patch-based optimal denoisers [12, 19] and link it to the observed linearity of diffusion models [17, 29, 30]. Unlike the optimal denoiser in eq. (3), trained diffusion models exhibit a "pass-through" behavior, retaining input information along high signal-to-noise (SNR) ratio data directions. We hypothesize that by constraining the locality in patch-based optimal denoisers, previous works effectively adopt the "pass-through" behavior from linear denoisers and thus are capable of producing novel images.

**Sampling voids.** The optimal denoiser $\hat{f}(x,t)$ is well-defined for all values of $x \in \mathbb{R}^n$ and $t \in (0,1)$, but the distribution of $(x_t, t)$ that a neural network denoiser is trained on has low-density regions that will be sparsely sampled throughout the training process. For example, when $t \to 0$ (low noise regime), regions in $\mathbb{R}^n$ far away from training data will be undersampled in training the diffusion model with score-matching objective in eq. (1). This is illustrated in Figure 1 (left), where the regions of small noise near the test images are not covered with any training samples. We will refer to the part of $\mathbb{R}^n \times [0,1]$ that is not covered by the empirical samples in eq. (1) as *sampling void* regions. On one hand, the behavior of the denoiser in those regions is critical for generalization. On the other hand, the optimal denoiser is not a good model of a trained diffusion model in these regions, as there were no empirical samples in this part of the space during training. In the next sections, we will build up intuition on how to reason about the behavior of the trained diffusion models in the sampling void regions.

**"Pass-through" denoisers.** When the optimal denoiser in eq. (3) is presented with a test image outside of the training dataset and the amount of noise is small, the softmax becomes more selective, and the optimal denoiser will predict $x_0$ to be the closest image in the dataset (see Figure 1, middle). This behavior prevents generalization, as *any* novel image will be "teleported" to its closest neighbor in the training dataset as $\sigma_t \to 0$.

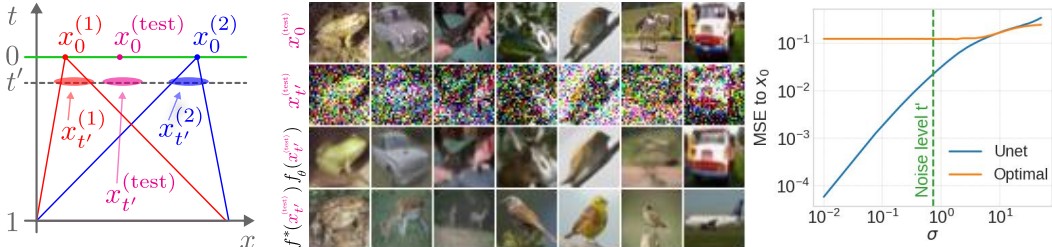

Figure 1: **Left**: We visualize the distribution of $x_t$ for two training data points $x_0^{(1)}$ and $x_0^{(2)}$ as high-probability-density "cones", as a function of spatial dimension $x$ and noise level $t$. Note how for a new testing point $x_0^{(test)}$ there exists noise level $t'$ such that noised versions of $x_{t'}^{(test)}$ are outside of any of the training "cones" and thus the behavior of the denoiser there is undefined. **Middle**: We take CIFAR10 test images (top) and add noise $\epsilon_{t'}$ (2nd row). With a single denoising step, a trained diffusion model $f_\theta$ "passes through" most of the coarse structure of the input image, and thus the output image is visually similar to the input (3rd row). Optimal denoiser $f^*$ instead "teleports" the image to the closest data point in the training dataset (4th row). **Right**: We compare MSE error of single-step denoising of $f_\theta$ (U-Net) and $f^*$ (Optimal). At low noise levels, $f_\theta$ removes noise from $x_{t'}^{(test)}$ but $f^*$ predicts a different image from $x_0^{(test)}$. At high noise levels, the outputs of $f_\theta$ and $f^*$ are similar.

Inspired by the observations in Figure 1, we develop an intuition about "pass-through" properties of denoisers. For small noise levels, a lot of information in the image is not destroyed by the added noise. It is natural to assume that a "good" denoiser, unlike the optimal denoiser, will retain this information in its estimation of $x_0$. *Which part of $x_t$ is not affected by a small amount of noise?* Informally, for natural images, low frequencies "survive" after adding small amounts of noise. We formalize this intuition by observing that 1) adding noise preserves the higher principal components of data, and 2) these principal components correspond to low-frequency features in natural images.

The principal components of the data come from the eigendecomposition of the covariance matrix $\text{Cov}(X) = U\text{diag}(\lambda_1^2, \lambda_2^2, \ldots \lambda_N^2)U^T$, where $U$ is a matrix of the eigenvectors and $\lambda_i^2$ are eigenvalues. The noise's covariance matrix $\sigma_t^2 I$ is also isotropic in this basis, so that the signal-to-noise ratio (SNR) along the $i$-th principal component is $\lambda_i^2/\sigma_t^2$. It is well-known in classical image processing literature [10] that for natural images, this eigenbasis approximates the Fourier basis, and thus the highest variance components (equiv. high SNR) correspond to low-frequency features. Thus, intuitively, the "pass-through" projection resembles a low-pass filter. Generally, however, it is not the case, and for more specific datasets, the "pass-through" projection is not just a low-pass filter. For instance, as we will show in Section 4, for datasets such as centered and normalized human faces (i.e., Celeba-HQ/FFHQ), the eigen-basis is very different from a Fourier basis, and thus the locality patterns observed in trained denoisers are not translation equivariant, nor isotropic.

**Connection to Gaussian data and linear denoiser.** The intuition in the previous paragraphs was built on an assumption that the dataset is well-described with a single covariance matrix (assuming Gaussian data distribution). In this case, one can craft a simple denoiser by just projecting the input noise images to their high-SNR principal components:

$$W_t = \frac{1}{\sqrt{\alpha_t}}U\text{diag}\left(\frac{\lambda_i^2}{\lambda_i^2 + \sigma_t^2}\right)U^T. \tag{6}$$

This denoiser, optimal under a Gaussian dataset assumption $x_0 \sim N(0, \Sigma)$ [15], is known as the Wiener filter [31]. At the same time, the Wiener filter is also the optimal linear denoiser minimizing eq. (1) under a linear constraint $f(x_t) = A_t x_t$ [31]. As we can see from eq. (6), it projects its input to the data's principal components, shrinks these projections according to their SNR, and projects them back to the data space. As reported in multiple previous works [17, 29], trained diffusion models exhibit linear behavior and can be surprisingly well-approximated with a Wiener filter. Later in this work, we extend these observations and demonstrate that the Wiener filter performs on par with or better than existing patch-based analytical denoisers [12, 18].

**Locality and sensitivity.** So far, we built up the intuition that a "good" denoiser should "pass-through" high-SNR components of the input in the *sampling void* regions—parts of space where optimal denoiser analysis is no longer effective, but critical for generalization. According to [12, 19],

locality of the denoisers' sensitivity fields plays a crucial role for its generalization. But how does it relate to the "pass-through" intuition above?

To show the relationship, we return to the notion of locality. By locality, we mean a limited, typically compact sensitivity field of a neural network. Formally, the sensitivity field of a denoiser $f(x, t)$ is its input–output Jacobian $S_f(x, t) = \partial f(x, t)/\partial x$. Kamb and Ganguli [12] approximate learned locality patterns in diffusion models with square patches, assuming them to be compact, roughly isotropic, and constant with respect to both the output pixel position and the input image. As we will demonstrate in Section 4, none of these assumptions is universal, and while they are reasonable for diverse datasets of natural images, the sensitivities of neural diffusers on more specialized datasets can violate any of the assumptions above.

For our analysis, we retain the assumption of independence of the model's sensitivity field with respect to the input image $x$ and lift all other assumptions, allowing locality patterns to take arbitrary shapes and depend on the output pixel. Under the assumption that the sensitivity $S_f(x, t)$ is constant w.r.t. $x$, the denoiser is linear in $x$ and takes the form $f(x, t) = A_t x + B_t$, where $A_t$ is the sensitivity and $B_t$ is a bias term. Recall that a solution to eq. (1) under a linear constraint is the Wiener filter and thus $A_t = W_t$ and $B_t = 0$ assuming the dataset is centered. For each individual output pixel $q$, the sensitivity takes the form:

$$S_f^q(x, t) = W_t^q = \frac{1}{\sqrt{\alpha_t}} \left[ U \mathrm{diag}\left( \frac{\mathrm{SNR}_i}{\mathrm{SNR}_i + 1} \right) U^T \right]^q \quad \text{where } \mathrm{SNR}_i = \frac{\lambda_i^2}{\sigma_t^2}. \tag{7}$$

In other words, with the assumption that locality patterns are shared for all input images, the sensitivity of the denoiser is identical to the high-SNR projection operator to the principal components of the covariance matrix. A key observation about the form of eq. (7) is that as $\sigma_t \to 0$ the signal-to-noise ratio for each component $\mathrm{SNR}_i \to \inf$ and thus $S_f^q(x, t) \to \mathbb{1}_q$, the indicator function at pixel $q$; i.e. the sensitivity field of the locally linear denoiser shrinks with smaller noise levels.

Under the local linearity assumption, the exact shape of those sensitivity fields is a function of the data and not of the model's architecture. In the subsequent sections, we empirically show that different architectures of denoising diffusion models learn sensitivity fields similar to those of linear denoisers, effectively approximating the projection operator to the high-SNR data's principal components.

Recall that previous work [12, 19] adopts the learned sensitivity fields from the trained diffusion networks, which play a crucial role in their generalization. As we reveal the connection of the sensitivity field to the local data statistics, we obtain valuable insights into the performance of the analytical models, especially for specialized datasets, where the data's principal components are far from being local and equivariant.

## 4 Validation

In this section, we perform extensive validation to support our claims. As the main backbone for a diffusion model, we use the DDPM U-Net [8] with removed self-attention to follow the protocol of [12]. As we show in Appendix B.2, removing the self-attention does not drastically affect performance, and our insights are valid for both architectures.

We begin by showing that different neural network architectures learn similar sensitivity fields, which in turn match projection operators to high-SNR principal components, or equivalently, the Wiener filter. We continue by demonstrating that the learned sensitivity fields are a property of the dataset and thus by manipulating the statistics of the data, we can force the diffusion model's sensitivity to take any shape, including being nonlocal for low noise levels.

Finally, to support our claim that the locality properties of trained diffusion models come from data statistics, we suggest a simple modification to previous patch-based analytical models [12, 19]. Instead of measuring locality from trained diffusion models, we limit the analytical model to only use high-SNR principal components. We benchmark this modification on five datasets and show that while being more interpretable, this algorithm also explains trained diffusion models better than other baselines. Additionally, we ablate our model and present comparisons in Appendix B.1.

**Locality pattern is shared across architectures.**    We compare the locality patterns throughout the denoising process across architectures (U-Net [8] and diffusion transformer (DiT) [21]) trained on

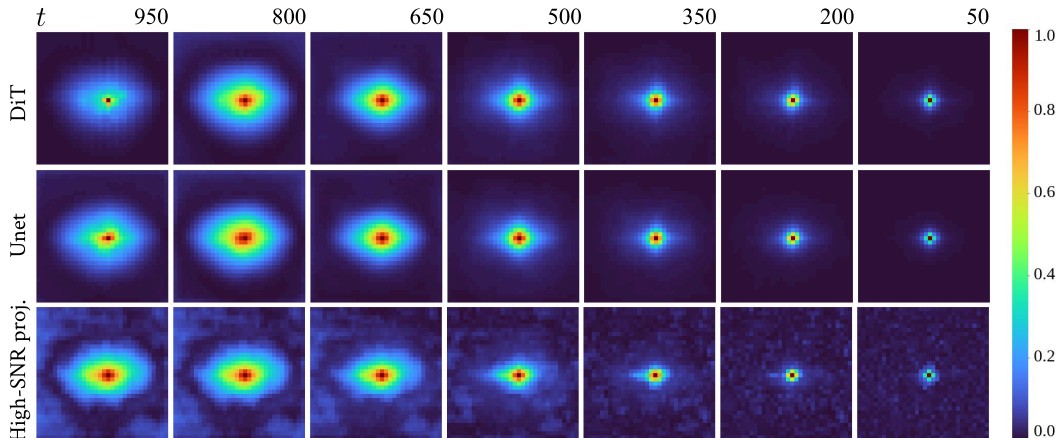

Figure 2: Comparison of sensitivity fields of deep denoisers and the projection operators to high-SNR data's components (i.e. the Wiener filter) on CIFAR-10 dataset. Sensitivity is measured at the center pixel w.r.t. $x_0$ prediction and throughout a 1000-step DDIM denoising process. Each image is averaged across 32 samples and normalized to [0,1].

the CIFAR10 dataset. Although there is architectural bias for locality in U-Nets due to convolutional layers, the self-attention layers in DiTs are global in scope, where every patch can attend to any other patch. Nevertheless, Figure 2 shows that U-Nets and DiTs share similar sensitivity fields. Surprisingly, these fields are similar to the shapes of high-SNR projector operators, i.e., the sensitivity of a Wiener filter. This provides evidence that the main reason for the diffusion models to exhibit locality properties is the correlation of the pixels between the images in the dataset. Our observation is that although the neural network architectural choices are important to accurately capture data statistics, they are not the main cause of locality patterns in diffusion models.

**Learned sensitivity fields are not always equivariant.** As we saw in the previous experiment, the sensitivity fields learned by diffusion models are similar across architectures and align with the data's principal components. In Figure 2, these fields appear roughly isotropic and equivariant—meaning the sensitivity $s^q(x)$ has the same shape for each pixel $q$, up to translation. This form of sensitivity is well-captured by the square-shaped patches in [12].

This behavior arises due to the high correlation among neighboring pixels and the translation equivariance inherent in the CIFAR10 dataset, which consists of diverse natural images.

However, for more specialized datasets, the principal components—and consequently the learned sensitivity fields—take on drastically different shapes. In particular, CelebA-HQ is a dataset of uniformly-scaled and centered human faces. The lack of translation equivariance and the unique pixel correlation patterns result in structured, location-dependent sensitivity fields.

In Figure 3, we observe the complex structure that arises in the sensitivity fields of a diffusion model trained on CelebA-HQ. Notably, the pattern of sensitivity is now highly dependent on the pixel's location. This experiment highlights the need for more flexible representations of sensitivity fields, especially for specialized datasets, than those in prior work.

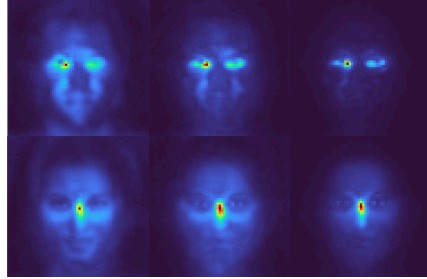

Figure 3: Average sensitivity fields of a trained DDPM on the CelebA-HQ dataset. The top row corresponds to an output pixel located near the left eye; the bottom row corresponds to an output pixel near the image center. Left to right: different noise levels corresponding to $t$ of 600, 400, 200.

**Manipulating the sensitivity field.** We show that by editing the dataset's statistics, we can manipulate the sensitivity fields of neural denoisers and make them take on any shape. With this manipulation,

U-Net-based denoisers can learn sensitivity fields that are not local, thus suggesting that the locality properties of learned denoisers emerge from dataset statistics.

In our experiment on CIFAR-10, we generate a modified dataset:

$$\hat{x}_0 = x_0 + \gamma cs, \qquad c \sim \text{Uniform}([-1, 1]^3),$$

where $x_0 \in \mathbb{R}^d$ is a training image, $s \in \{0, 1\}^d$ is a fixed binary mask in the shape of the letter "W", and $c \in \mathbb{R}^3$ is a random RGB vector (single color per image) with $\gamma > 0$ controlling signal strength. Note that this transformation does not change the first-order moments of the data as $\mathbb{E}[\gamma cs] = 0$.

With this, we train a new DDPM U-Net from scratch on the modified CIFAR10 dataset and then study its sensitivity fields. Let $\lambda_w$ be the variance of the perturbation. We choose $\gamma$ so that $\lambda_w \approx \sigma_{t_\star}$ for some intermediate $t_\star$, i.e., the variance of the added signal matches the variance of the noise. As we can see[2] from Figure 4, the "W"-shaped sensitivity field emerges for all $t$ where $\sigma_t \ll \lambda_w$. Crucially, this demonstrates that *any* desired pattern can be induced in the sensitivity of a trained neural network by embedding the pattern into the data covariance.

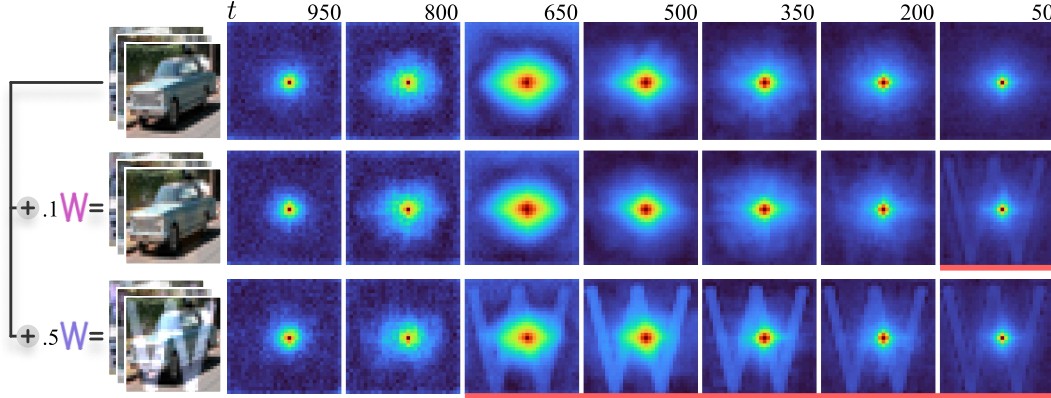

Figure 4: We slightly manipulate pixels' correlations across the CIFAR-10 dataset such that a desired pattern emerges in the sensitivity of a trained diffusion model. In particular, a DDPM diffusion model trained on the CIFAR-10 dataset (sample on the top left) has a coarse-to-fine sensitivity field (top row, noise level decreases from left to right). For each image in the dataset, we edit pixel correlations by adding the desired pattern with random color and weights $\gamma = 0.1$ (middle row) and $\gamma = 0.5$ (bottom row). DDPM models trained on those manipulated datasets exhibit the pattern in their sensitivity fields. We underscore the time-steps for which $SNR_W > 0.1$, i.e. $\lambda_W^2 > 0.1\sigma_t^2$. This supports our claim that the locality in diffusion models arises not from the inductive bias (i.e. usage of convolutional layers) but from the data statistics.

**Our model.** Previous work uses rectangular binary masks fitted to the sensitivity fields of trained denoisers. In this paper, we demonstrated that the sensitivity fields of the trained denoisers emerge from the data covariance. We consider constant sensitivity fields, i.e., $A_t^q$ is not a function of $x$. If the sensitivity field is constant, the denoiser is linear, and we know that the optimal linear denoiser is the Wiener filter. Using this intuition, we consider a generalized notion of locality and show that the locality property is equivalent to a subspace projection, which can be written as an orthogonal change of basis followed by a masking operator. We provide a detailed derivation in Appendix A.3.

Replacing the measured sensitivity field as in [12, 19] with the projection operator to high-SNR components (sensitivity of the optimal linear denoiser) performs on par with or better than the patch-based optimal denoisers. More formally, we suggest using the following analytical model:

$$\hat{f}(x, t) = \sum_{i=1}^{N} w_i(x, t)x_0^i, \quad w_i^q(x) = \underset{i}{\text{softmax}} \left\{ -\frac{1}{2\sigma_t^2} \left\| \hat{W}_t^q \left( \frac{1}{\sqrt{\alpha_t}} x - x_0^j \right) \right\|_2^2 \right\}_{j \in [N]}, \quad (8)$$

where $\hat{W}_t^q$ is a $q$-th row of the Wiener matrix binarized with a threshold $\tau = 0.02$ (we are using $\tau = 0.02$ relative to the max value in the row unless stated otherwise). For MNIST and Fashion

---

[2]To aid visualization, we apply the square root to the sensitivity field and plot it with "turbo" color map. In the rest of the paper, the color map is applied to the raw signal.

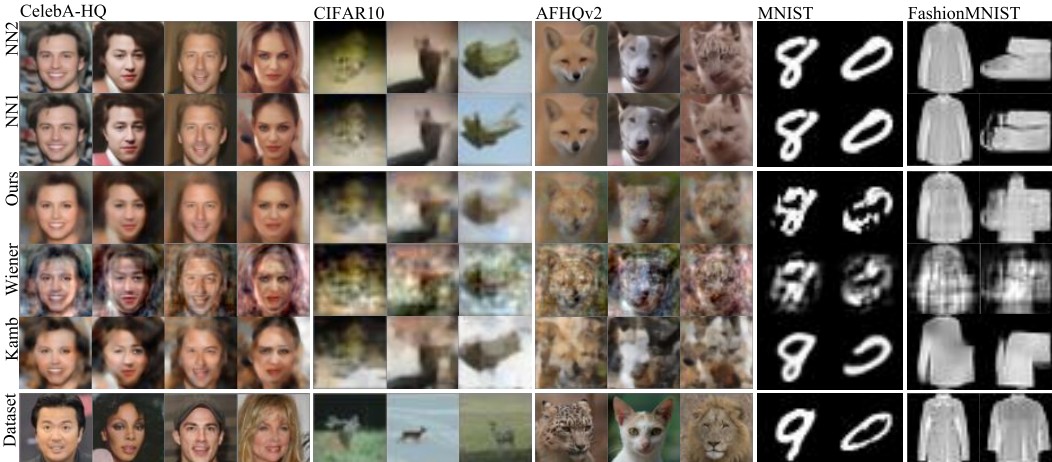

Figure 5: **Qualitative comparison**. In this figure, we compare our analytical model (3rd row) with multiple baselines: Wiener filter (4th row), Kamb and Ganguli [12] analytical model (5th row). All images are generated with the same initial noise sample with 10 steps of DDIM [27]. In the top row, we provide the results of generation with two trained neural networks, NN1 and NN2 – both are instances of the same DDPM U-Net [8], but trained with different seeds. The distance in Table 1 is measured with respect to NN1. In the last row we provide the nearest image from the dataset for our final generation w.r.t. L2 distance.

MNIST we use $\tau = 0.005$. We provide a detailed derivation of this formula and the ablation of $\tau$ in Appendices A.2 and B.1. Key differences of our model compared to [12] and [19] are:

1. **Enhanced interpretability.** Instead of fitting patch sizes and/or shapes to trained models, we obtain them analytically from dataset statistics.
2. **No equivariance.** While prior work claims equivariance as an important property of diffusion models [12, 18], we instead find that introducing the translation group integral to Equation (8) does not improve performance but increases inference time.
3. **Locality specific to each pixel in the image**. Unlike prior work, we do not enforce the shape of the patches to be shared across all pixels of the image, instead relying on the dataset's statistics. This is particularly important for datasets with nonlocal covariances, such as datasets of faces.

Our model is nonlinear and does not assume that the dataset is Gaussian. Rather, it approximates only the locality fields with the second-order statistics of the dataset. We benchmark our analytical model by measuring the $r^2$-coefficient of determination and mean-squared error (MSE) between the predictions of the analytical model and a trained DDPM [8] given the same starting noise.

For comparison, we chose five datasets with diverse sets of statistics: CIFAR10 [16], a dataset of diverse $32 \times 32$ natural images; CelebA-HQ [13] and AFHQv2 [3], datasets of centered faces and animals in $64 \times 64$; and MNIST [6] and FashionMNIST [32], datasets of binary centered images in $28 \times 28$ resolution. We compare against the vanilla optimal denoiser in eq. (3), the Wiener filter [31], and the patch-based optimal denoising algorithm by Kamb and Ganguli [12]. Additionally, to capture the variance in signal-to-noise mapping of diffusion models, we compare with another trained diffusion model with the same architecture and dataset, but with different weight initialization.

Table 1 and Figure 5 show that our analytical model outperforms all of the baselines, with the Wiener filter being almost always in second place. Kamb and Ganguli's model qualitatively performs worse on CelebA-HQ due to patch-based locality erasing eyes and blurring out facial features, while other models using dataset-dependent locality retain these features. This experiment confirms our hypothesis that the modeling of correct locality in the analytical models is key to explaining trained diffusion models and that those localities come from the dataset statistics. We provide additional quantitative results for AFHQv2 and Fashion-MNIST in Appendix B.7.

Table 1: We provide a quantitative comparison that measures how well each analytical model explains the trained image diffusion models. All metrics are averaged over 128 samples. Best results are highlighted in green and second best in maroon.

| Method | CIFAR10 | | CelebA-HQ | | MNIST | |
|---|---|---|---|---|---|---|
| | $r^2 \uparrow$ | MSE$\downarrow$ | $r^2 \uparrow$ | MSE$\downarrow$ | $r^2 \uparrow$ | MSE$\downarrow$ |
| Optimal | -0.549±0.774 | 0.191±0.044 | 0.400±0.298 | 0.101±0.023 | 0.187±0.204 | 0.231±0.036 |
| Wiener (linear) | 0.408±0.092 | 0.032±0.004 | 0.818±0.039 | 0.031±0.004 | 0.469±0.066 | 0.161±0.014 |
| Kamb [12] | 0.303±0.126 | 0.065±0.017 | 0.831±0.073 | 0.028±0.005 | 0.402±0.092 | 0.188±0.039 |
| **Ours** | 0.589±0.078 | 0.028±0.008 | 0.902±0.032 | 0.016±0.006 | 0.491±0.051 | 0.153±0.015 |
| Another DDPM | 0.852±0.113 | 0.023±0.002 | 0.981±0.007 | 0.004±0.001 | 0.969±0.082 | 0.007±0.019 |

# 5 Conclusion, Limitations, and Future Work

In this work, we demonstrated that locality in diffusion models emerges from dataset statistics rather than architectural inductive biases. Through theoretical analysis and empirical validation, we showed that both U-Nets and Transformers learn sensitivity fields that closely align with projections onto high-SNR principal components of the training data. This intuition links the behavior of the diffusion models to linear denoisers, or equivalently, the Wiener filter. Using our theoretical insights, we show that by editing the dataset's statistics, we are able to manipulate the sensitivity fields of trained diffusion models and can make them take arbitrary shapes, including highly nonlocal ones. Finally, our analytical model, based on dataset statistics, outperforms previous approaches in approximating trained diffusion models across multiple datasets.

This work addresses a critical gap in understanding how diffusion models generalize rather than memorize, highlighting the "pass-through" behavior where high-SNR components are preserved. Limitations of our approach include a focus on simpler architectures and reliance on second-order statistics, while deep diffusion networks can capture higher-order statistics of the data. In particular, we are making a strong assumption that the locality fields are constant with respect to the input images. Studying non-linear regimes of the neural diffusers can deepen our understanding of the mechanisms of image generation. Future work on complex architectures, higher-order statistics, and conditional generation has the potential to further explain the theory-practice gap in diffusion models.

## Acknowledgments

The authors thank Christopher Scarvelis, Ishaan Chandratreya and Vitor Guizilini for their thoughtful insights and feedback.

Vincent Sitzmann was supported by the National Science Foundation under Grant No. 2211259, by the Singapore DSTA under DST00OECI20300823 (New Representations for Vision, 3D Self-Supervised Learning for Label-Efficient Vision), by the Intelligence Advanced Research Projects Activity (IARPA) via Department of Interior/ Interior Business Center (DOI/IBC) under 140D0423C0075, by the Amazon Science Hub, and by the MIT-Google Program for Computing Innovation.

The MIT Geometric Data Processing Group acknowledges the generous support of Army Research Office grant W911NF2110293, of National Science Foundation grants IIS2335492 and OAC2403239, from the CSAIL Future of Data and FinTechAI programs, from the MIT–IBM Watson AI Laboratory, from the Wistron Corporation, from the MIT Generative AI Impact Consortium, from the Toyota–CSAIL Joint Research Center, and from Schmidt Sciences.

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

# Part I

# Appendix

## Table of Contents

# A  Derivations and proofs

In this section, we provide detailed derivations and proofs for the background and the claims made in the paper.

## A.1  Optimal denoiser: derivation, equivalence of $\epsilon$ and $x_0$ parametrization

We begin by defining the optimal denoiser for the $x_0$ parameterization we use in the paper.

**Definition A.1.** *The optimal denoiser $\hat{f}(x,t)$ for a data distribution $X$ at a particular noise level $t$ is the minimizer of the loss function*

$$\min_{f} \; \mathop{\mathbb{E}}_{\substack{x_0 \sim X \\ \epsilon \sim N(0,I)}} \left\| f(\sqrt{\alpha_t}x_0 + \sqrt{1-\alpha_t}\epsilon, t) - x_0 \right\|_2^2 \tag{9}$$

Recall that $\sigma_t^2 = \frac{1-\alpha_t}{\alpha_t}$.

**Proposition A.2.** *When $X = \left\{x_0^i\right\}_{i \in [N]}$ is a finite empirical distribution, the optimal denoiser $\hat{f}(x,t)$ has the following analytical expression:*

$$\hat{f}(x,t) = \sum_i x_0^i \mathop{\mathrm{softmax}}_i \left\{ -\frac{1}{2\sigma_t^2} \left\| \frac{x}{\sqrt{\alpha_t}} - x_0^j \right\|^2 \right\}. \tag{10}$$

*Proof.* We first write down the objective (9) in terms of the random variable $x = \sqrt{\alpha_t}x_0 + \sqrt{1-\alpha_t}\epsilon$:

$$\mathop{\mathbb{E}}_{\substack{x_0 \sim X \\ \epsilon \sim N(0,I)}} \left[ \left\| f(\sqrt{\alpha_t}x_0 + \sqrt{1-\alpha_t}\epsilon, t) - x_0 \right\|_2^2 \right]$$

$$= \mathop{\mathbb{E}}_{\substack{x_0 \sim X \\ x \sim N(\sqrt{\alpha_t}x_0, (1-\alpha_t)I)}} \left\| f(x,t) - x_0 \right\|_2^2$$

$$= \int \mathop{\mathbb{E}}_{x_0 \sim X} \left[ \left( \sqrt{2\pi}(1-\alpha_t) \right)^{-n} \exp\left( -\left\| x_0 - \frac{x}{\sqrt{\alpha_t}} \right\|^2 / 2\sigma_t^2 \right) \left\| f(x,t) - x_0 \right\|_2^2 \right] dx$$

We then minimize the integral coordinate-wise for each $x$ to get the optimal $f(x,t)$:

$$0 = \mathop{\mathbb{E}}_{x_0 \sim X} \left[ \exp\left( -\left\| x_0 - \frac{x}{\sqrt{\alpha_t}} \right\|^2 / 2\sigma_t^2 \right) (\hat{f}(x,t) - x_0) \right]$$

$$\hat{f}(x,t) = \frac{\sum_i x_0^i \exp(-\left\| \frac{x}{\sqrt{\alpha_t}} - x_0^i \right\|^2 / 2\sigma_t^2)}{\sum_j \exp(-\left\| \frac{x}{\sqrt{\alpha_t}} - x_0^j \right\|^2 / 2\sigma_t^2)}.$$

Using the definition of $\mathrm{softmax}_i \left\{a_j\right\}_{j \in [N]} = \frac{\exp(a_i)}{\sum_{j=1}^N \exp(a_j)}$, we get (10).

$\square$

**Definition A.3.** *The optimal denoiser $\hat{\epsilon}(z,t)$ for a data distribution $X$ at a particular noise level $t$ is the minimizer of the loss function*

$$\min_{f} \; \mathop{\mathbb{E}}_{\substack{x_0 \sim X \\ \epsilon \sim N(0,I)}} \left\| f(\sqrt{\alpha_t}x_0 + \sqrt{1-\alpha_t}\epsilon, t) - \epsilon \right\|_2^2 \tag{11}$$

**Proposition A.4.** *When $X = \left\{x_0^i\right\}_{i \in [N]}$ is a finite empirical distribution, the optimal denoiser for the $\epsilon$-parameterization $\hat{\epsilon}(x,t)$ can be written in terms of that of the $x$-parameterization $\hat{f}(x,t)$:*

$$\hat{\epsilon}(x,t) = \frac{x - \sqrt{\alpha_t}\hat{f}(x,t)}{\sqrt{1-\alpha_t}} \tag{12}$$

*Proof.* We follow the same proof as Proposition A.2, with the main difference being the following step:

$$\mathbb{E}_{\substack{x_0 \sim X \\ \epsilon \sim N(0,I)}} \left[ \left\| f(\sqrt{\alpha_t}x_0 + \sqrt{1-\alpha_t}\epsilon, t) - \epsilon \right\|_2^2 \right]$$

$$= \mathbb{E}_{\substack{x_0 \sim X \\ x \sim N(\sqrt{\alpha_t}x_0, (1-\alpha_t)I)}} \left\| f(x,t) - \frac{x - \sqrt{\alpha_t}x_0}{\sqrt{1-\alpha_t}} \right\|_2^2.$$

$\square$

**Remark A.5.** *Another way to prove Proposition A.2 and Proposition A.4 is to show that the optimal solutions are of the form $\mathbb{E}[x_0 \mid x]$ and $\mathbb{E}[\epsilon \mid x]$, where $x \sim N(\sqrt{\alpha_t}x_0, (1-\alpha_t)I)$. Then it becomes clear that the two expressions are linearly related to each other.*

## A.2 Patch-based optimal denoiser: formal derivation

We now turn to the patch-based denoiser, incorporating both locality and equivariance constraints into the optimal denoising problem as suggested in [12], repeating the derivations in the notations of this manuscript. Let $X = \{x_0^i\}_{i=1}^N$ be a finite empirical distribution of images, and let

$$M_t^q : \mathbb{R}^{d \times d} \to \mathbb{R}^{d \times d}$$

denote the operator that masks out a $p \times p$ patch centered at pixel $q$, setting the rest of the pixels to 0.

As suggested in [12], we impose two constraints on each patch-wise function $f^q$:

1. *Locality:*

$$f^q(x,t) = f^q\big(M_t^q x, t\big).$$

2. *Equivariance:* For every 2D translation $g \in T(2)$,

$$f\big(g \circ x, t\big) = g \circ f(x,t), \quad \implies \quad f^q\big(g \circ x, t\big) = f^{g^{-1}q}(x,t),$$

i.e. denoising commutes with the action of $T(2)$ and relocates patches accordingly.

**Definition A.6.** *The patch-based optimal denoiser $\hat{f}(x,t)$ for a data distribution $X$ at a particular noise level $t$ is the minimizer of the loss function*

$$\min_f \; \mathbb{E}_{x_0 \sim X, \, \epsilon \sim N(0,I), \, t \sim [0,1]} \big\| f\big(\sqrt{\alpha_t}\, x_0 + \sqrt{1-\alpha_t}\, \epsilon, \, t\big) - x_0 \big\|_2^2$$
$$\text{s.t. } f^q(x,t) = f^q\big(M_t^q x, t\big), \quad q = 1, \ldots, Q, \qquad \text{(locality)} \qquad (13)$$
$$f\big(g \circ x, t\big) = g \circ f(x,t), \quad \forall g \in T(2). \qquad \text{(equivariance)}$$

**Proposition A.7** (Patch-based optimal denoiser). *Under the empirical distribution $X = \{x_0^i\}_{i=1}^N$, the minimizer $\{\hat{f}^q\}$ of eq. (13) is given, for each patch location $q$, by*

$$\hat{f}^q(x,t) = \sum_i \sum_{g \in T(2)} \big(g \circ x_0^i\big)^q \, \text{softmax}_i \left( -\frac{1}{2\sigma_t^2} \left\| M_t^q \left( \frac{x}{\sqrt{\alpha_t}} - g \circ x_0^i \right) \right\|^2 \right), \qquad (14)$$

*and the full-image denoiser is obtained by reconstructing the final image from the pixels above.*

*Proof of Patch-based optimal denoiser.* We prove this result in three steps: (1) decomposition into per-pixel optimization, (2) equivalence of equivariance constraint and data augmentation, and (3) derivation of the local form.

**Step 1: Decomposition into per-pixel optimization.** Let $P_q : \mathbb{R}^{d \times d} \to \mathbb{R}$ denote the operator that extracts pixel $q$ from an image, and define $H = \sum_{q=1}^Q P_q P_q^T$ where $P_q^T$ places a scalar value at pixel $q$ and zeros elsewhere. Since pixels are disjoint, $P_i P_j^T = 0$ for $i \neq j$, making $\{P_q P_q^T\}$ orthogonal projections with $\sum_{q=1}^Q P_q P_q^T = I$.

By orthogonality of pixel projections:

$$\|f(x) - x_0\|_2^2 = \left\|\sum_{q=1}^{Q} P_q P_q^T (f(x) - x_0)\right\|_2^2$$

$$= \sum_{q=1}^{Q} \|P_q f(x) - P_q x_0\|_2^2$$

$$= \sum_{q=1}^{Q} |f^q(x) - x_0^q|^2$$

Therefore, the original minimization problem decomposes as:

$$\min_{f} \mathbb{E}_{x_0 \sim X, \epsilon \sim N(0,I)} \|f(\sqrt{\alpha_t}x_0 + \sqrt{1-\alpha_t}\epsilon) - x_0\|_2^2$$

$$= \sum_{q=1}^{Q} \min_{f^q} \mathbb{E}_{x_0 \sim X, \epsilon \sim N(0,I)} \left[f^q(\sqrt{\alpha_t}x_0 + \sqrt{1-\alpha_t}\epsilon) - x_0^q\right]^2$$

Each pixel can be optimized independently.

**Step 2: Equivariance constraint equals data augmentation.** For the $q$-th pixel problem with equivariance constraint:

$$\min_{f^q} \quad \mathbb{E}_{x_0 \sim X, \epsilon \sim N(0,I)} \left[f^q(\sqrt{\alpha_t}x_0 + \sqrt{1-\alpha_t}\epsilon) - x_0^q\right]^2 \tag{15}$$
$$\text{s.t.} \quad f^q(g \circ x) = (g \circ f(x))^q \quad \forall g \in T(2)$$

The equivariance constraint implies that for any translation $g$: $f^q(g \circ x) = f^{g^{-1}q}(x)$

Now consider the data-augmented problem:

$$\min_{f^q} \mathbb{E}_{x_0 \sim X, \epsilon \sim N(0,I)} \mathbb{E}_{g \sim T(2)} \left[f^q(g \circ (\sqrt{\alpha_t}x_0 + \sqrt{1-\alpha_t}\epsilon)) - (g \circ x_0)^q\right]^2 \tag{16}$$

We want to prove that (15) is equivalent to (16). To do so, we first show that the optimal solution of (15) does not change with data-augmentation, then show that the optimal solution of (16) satisfies the equivariance constraint.

**Constrained optima invariant under data-augmentation** Let $x = \sqrt{\alpha_t}x_0 + \sqrt{1-\alpha_t}\epsilon$. Since translation commutes with noise addition and $(g \circ x_0)^q = x_0^{g^{-1}q}$, (15) under data-augmentation becomes:

$$\min_{f^q} \mathbb{E}_{x_0 \sim X, \epsilon \sim N(0,I)} \mathbb{E}_{g \sim T(2)} \left[f^q(g \circ x) - x_0^{g^{-1}q}\right]^2$$
$$\text{s.t.} \ f^q(g \circ x) = (g \circ f(x))^q \quad \forall g \in T(2)$$

If $f^q$ satisfies the equivariance constraint, then $f^q(g \circ x) = f^{g^{-1}q}(x)$, so:

$$\arg\min_{f^q} \mathbb{E}_{x_0 \sim X, \epsilon \sim N(0,I), g \sim T(2)} \left(f^q(g \circ x) - x_0^{g^{-1}q}\right)^2 = \arg\min_{f^q} \mathbb{E}_{x_0, \epsilon, g} \left(f^{g^{-1}q}(x) - x_0^{g^{-1}q}\right)^2$$

$$= \arg\min_{f^q} \mathbb{E}_{x_0, \epsilon} \sum_{r=1}^{Q} (f^r(x) - x_0^r)^2$$

$$= \arg\min_{f^q} \mathbb{E}_{x_0, \epsilon} \left(f^q(x) - x_0^q\right)^2$$

**Data-augmented optima is equivariant** We can solve the data-augmented problem (16) and check that its optimal solution $\hat{f}^q$ is equivariant under the transformation $x \to h \circ x$, for any $h \in T(2)$:

$$
\begin{aligned}
\hat{f}^q(h \circ x) &= \sum_{i=1}^{N} \sum_{g \in T(2)} (g \circ x_0^i)^q \cdot \mathrm{softmax}_{i,g} \left\{ -\frac{1}{2\sigma_t^2} \left\| \frac{1}{\sqrt{\alpha_t}} h \circ x - g \circ x_0^i \right\|^2 \right\} \\
&= \sum_{i=1}^{N} \sum_{g \in T(2)} (g \circ x_0^i)^q \cdot \mathrm{softmax}_{i,g} \left\{ -\frac{1}{2\sigma_t^2} \left\| \frac{x}{\sqrt{\alpha_t}} - h^{-1} \circ g \circ x_0^i \right\|^2 \right\} \\
&= \sum_{i=1}^{N} \sum_{g' \in T(2)} (h \circ g' \circ x_0^i)^q \cdot \mathrm{softmax}_{i,g'} \left\{ -\frac{1}{2\sigma_t^2} \left\| \frac{x}{\sqrt{\alpha_t}} - g' \circ x_0^i \right\|^2 \right\} \\
&= (h \circ f(x))^q,
\end{aligned}
$$

where $g' = h^{-1} \circ g$.

**Step 3: Local form derivation.** From the data augmentation equivalence, the optimal denoiser for pixel $q$ minimizes:

$$
\mathbb{E}_{x_0 \sim X, \epsilon \sim N(0,I)} \mathbb{E}_{g \sim T(2)} \left[ f^q(\sqrt{\alpha_t}(g \circ x_0) + \sqrt{1 - \alpha_t}\epsilon) - (g \circ x_0)^q \right]^2
$$

Let $x = \sqrt{\alpha_t} g \circ x_0 + \sqrt{1 - \alpha_t}\epsilon$. Writing the objective as an integral over $x$, we get:

$$
\int \mathbb{E}_{x_0 \sim X, g \sim T(2)} (2\pi(1 - \alpha_t))^{-d} \exp \left( - \left\| g \circ x_0 - \frac{x}{\sqrt{\alpha_t}} \right\|^2 / 2\sigma_t^2 \right) (f^q(x) - (g \circ x_0)^q)^2 \; dx
$$

With the locality constraint $f^q(x) = f^q(M_t^q x)$, we can "integrate out" the coordinates of $x$ that are masked out by $M_t^q$ (as $f^q$ does not depend on them), to get:

$$
\int \mathbb{E}_{x_0 \sim X, g \sim T(2)} (2\pi(1 - \alpha_t))^{-p} \exp \left( - \left\| M_t^q(g \circ x_0 - \frac{x}{\sqrt{\alpha_t}}) \right\|^2 / 2\sigma_t^2 \right) (f^q(M_t^q x) - (g \circ x_0)^q)^2 \; d(M_t^q x)
$$

Solving for the optimal $\hat{f}^q$, we get:

$$
0 = \mathbb{E}_{\substack{x_0 \sim X \\ g \sim T(2) \\ \epsilon \sim N(0,I)}} \exp \left( - \frac{\| \frac{1}{\sqrt{\alpha_t}} M_t^q x - M_t^q(g \circ x_0) \|^2}{2\sigma_t^2} \right) (f^q(M_t^q x) - (g \circ x_0)^q)
$$

Rearranging:

$$
\hat{f}^q(M_t^q x) = \frac{\sum_{i=1}^{N} \sum_{g \in T(2)} (g \circ x_0^i)^q \exp \left( -\| \frac{1}{\sqrt{\alpha_t}} M_t^q x - M_t^q(g \circ x_0^i)\|^2 / 2\sigma_t^2 \right)}{\sum_{j=1}^{N} \sum_{h \in T(2)} \exp \left( -\| \frac{1}{\sqrt{\alpha_t}} M_t^q x - M_t^q(h \circ x_0^j)\|^2 / 2\sigma_t^2 \right)}
$$

Using the softmax notation:

$$
\hat{f}^q(x, t) = \sum_{i=1}^{N} \sum_{g \in T(2)} (g \circ x_0^i)^q \cdot \mathrm{softmax}_{i,g} \left\{ -\frac{1}{2\sigma_t^2} \left\| \frac{1}{\sqrt{\alpha_t}} M_t^q x - M_t^q(g \circ x_0^i) \right\|^2 \right\}
$$

This completes the proof. □

## A.3 Ours: why do we binarize the sensitivity field

In this section, we provide justification for our algorithm provided in the main paper. In particular, we formally justify why it makes sense to binarize the sensitivity fields into a mask of zeros and ones.

At first, we generalize the patch-based optimal denoiser by relaxing the locality constraint. Instead of restricting to patch extraction operators $M_t^q$, we consider linear operators $A_t^q : \mathbb{R}^{d \times d} \to \mathbb{R}^{d \times d}$ that can capture more complex spatial relationships.

**Definition A.8** (Generalized masked optimal denoiser). *The generalized masked optimal denoiser* $\hat{f}(x,t)$ *for a data distribution $X$ at noise level $t$ is the minimizer of:*

$$\min_f \mathop{\mathbb{E}}_{x_0 \sim X, \, \epsilon \sim N(0,I)} \left\| f\big(\sqrt{\alpha_t}\,x_0 + \sqrt{1-\alpha_t}\,\epsilon,\ t\big) - x_0 \right\|_2^2 \tag{17}$$

$$s.t.\ f^q(x,t) = f^q\big(A_t^q x,\ t\big), \quad q = 1, \dots, Q, \qquad \text{(generalized locality)}$$

$$f\big(g \,\circ\, x,\ t\big) = g \,\circ\, f(x,t), \quad \forall\, g \in T(2) \qquad \text{(equivariance)}$$

*where $A_t^q : \mathbb{R}^{d \times d} \to \mathbb{R}^{d \times d}$ is a linear operator.*

Note that when $A_t^q$ is invertible, $f^q$ is effectively unconstrained (i.e. there is no locality). On the other hand, when $A_t^q = 0$, $f^q$ does not depend on $x$. The following result interpolates between these two extreme cases, showing that the optimal denoiser depends only on the row-space $A_t^q$.

**Proposition A.9** (Generalized masked optimal denoiser). *Following the decomposition and data augmentation equivalence from the previous section, the optimal denoiser for pixel $q$ under the generalized locality constraint is:*

$$\hat{f}^q(x,t) = \sum_{i=1}^N \sum_{g \in T(2)} (g \circ x_0^i)^q \cdot softmax_{i,g} \left\{ -\frac{1}{2\sigma_t^2} \left\| P_t^q \left( \frac{1}{\sqrt{\alpha_t}} x - g \circ x_0^i \right) \right\|^2 \right\} \tag{18}$$

*where $P_t^q = (A_t^q)^T \left( A_t^q (A_t^q)^T \right)^\dagger A_t^q$ is the orthogonal projection matrix onto the row space of $A_t^q$.*

*Proof.* The objective (17) can be written as

$$\int \mathop{\mathbb{E}}_{x_0,g} (2\pi(1-\alpha_t))^{-d} \exp\left( - \left\| g \circ x_0 - \frac{1}{\sqrt{\alpha_t}} x \right\|^2 / 2\sigma_t^2 \right) (f^q(x) - (g \circ x_0)^q)^2 \, dx.$$

We use SVD to write $A_t^q = U^T \Lambda V$, and notice that $P_t^q = V\Lambda^\dagger V^T$, where $\Lambda^\dagger$ is a diagonal matrix with 0 or 1 diagonal entries depending if the corresponding diagonal entry of $\Lambda$ is nonzero. Then we impose the constraint $f^q(x) = f^q(A_t^q x)$ and perform a change of variables $y = Vx$.

$$\int \mathop{\mathbb{E}}_{x_0,g} (2\pi(1-\alpha_t))^{-d} \exp\left( - \left\| g \circ x_0 - \frac{1}{\sqrt{\alpha_t}} x \right\|^2 / 2\sigma_t^2 \right) (f^q(A_t^q x) - (g \circ x_0)^q)^2 \, dx$$

$$= \int \mathop{\mathbb{E}}_{x_0,g} (2\pi(1-\alpha_t))^{-d} \exp\left( - \left\| V(g \circ x_0) - \frac{1}{\sqrt{\alpha_t}} y \right\|^2 / 2\sigma_t^2 \right) (f^q(\Lambda y) - (g \circ x_0)^q)^2 \, dy,$$

where we have absorbed $U$ into $f^q$ as it is an invertible matrix. Next, we integrate out the coordinates of $y$ corresponding to the zero-values in $\Lambda$, and absorb non-zero entries of $\Lambda$ into $f^q$, to get

$$\int \mathop{\mathbb{E}}_{x_0,g} (2\pi(1-\alpha_t))^{-d} \exp\left( - \left\| \Lambda^\dagger \left( V(g \circ x_0) - \frac{1}{\sqrt{\alpha_t}} y \right)^2 / 2\sigma_t^2 \right\| \right) (f^q(\Lambda^\dagger y) - (g \circ x_0)^q)^2 \, dy.$$

We then solve for the optimal $\hat{f}^q(y)$ and change variables to get $\hat{f}^q(x)$:

$$\hat{f}^q(y,t) = \sum_{i=1}^N \sum_{g \in T(2)} (g \circ x_0^i)^q \cdot softmax_{i,g} \left\{ -\frac{1}{2\sigma_t^2} \left\| \frac{1}{\sqrt{\alpha_t}} \Lambda^\dagger \left( y - V(g \circ x_0^i) \right) \right\|^2 \right\}$$

$$\hat{f}^q(x,t) = \sum_{i=1}^N \sum_{g \in T(2)} (g \circ x_0^i)^q \cdot softmax_{i,g} \left\{ -\frac{1}{2\sigma_t^2} \left\| \frac{1}{\sqrt{\alpha_t}} \Lambda^\dagger V \left( x - g \circ x_0^i \right) \right\|^2 \right\}$$

$$= \sum_{i=1}^N \sum_{g \in T(2)} (g \circ x_0^i)^q \cdot softmax_{i,g} \left\{ -\frac{1}{2\sigma_t^2} \left\| P_t^q \left( \frac{1}{\sqrt{\alpha_t}} x - g \circ x_0^i \right) \right\|^2 \right\}.$$

$$\square$$

**Corollary A.9.1** (Diagonal operators and mask binarization). *When $A_t^q = diag(a_i^q)$ is further constrained to be diagonal matrix with entries $a_i^q$, the optimal denoiser simplifies to:*

$$\hat{f}^q(x,t) = \sum_{i=1}^{N} \sum_{g \in T(2)} (g \circ x_0^i)^q \cdot softmax_{i,g} \left\{ -\frac{1}{2\sigma_t^2} \left\| B^q \odot \left( \frac{1}{\sqrt{\alpha_t}} x - g \circ x_0^i \right) \right\|^2 \right\} \quad (19)$$

*where $B^q$ is the binary mask with $B_i^q = 1$ if $a_i^q \neq 0$ and $B_i^q = 0$ otherwise, and $\odot$ denotes element-wise multiplication.*

*Proof.* We begin with the generalized optimal denoiser from equation (18) and substitute the diagonal operator $A_t^q = diag(a_1^q, a_2^q, \ldots, a_Q^q)$. Then $P_t^q = (A_t^q)^T \left( A_t^q (A_t^q)^T \right)^{\dagger} A_t^q = B^q$. We get a binary mask because oefficients $a_j^q$ cancel out completely when $a_j^q \neq 0$. The actual values of non-zero $a_j^q$ do not affect the optimal denoiser—only whether $a_j^q = 0$ or $a_j^q \neq 0$ matters. Then we can apply Proposition A.9 to get

$$\hat{f}^q(x,t) = \sum_{i=1}^{N} \sum_{g \in T(2)} (g \circ x_0^i)^q \cdot softmax_{i,g} \left\{ -\frac{1}{2\sigma_t^2} \left\| B^q \odot \left( \frac{1}{\sqrt{\alpha_t}} x - g \circ x_0^i \right) \right\|^2 \right\}$$

This completes the proof, showing that the optimal denoiser depends only on the binary support of the diagonal operator, not on the specific non-zero values. $\square$

**Connection to patch-based denoiser:** When $a_j^q = 0$, the corresponding pixel $j$ is effectively removed from the optimization, as it contributes zero to the distance metric. This is precisely the locality constraint from the patch-based denoiser: pixels outside the patch (where $a_j^q = 0$) do not influence the denoising of pixel $q$.

**Remark A.10** (Justification for binary masks). *In summary, we showed in Proposition A.9 that the only interesting generalized locality matrices are binary masks. In particular, when the masking operator $A_t^q$ has a diagonal structure, the specific values of the non-zero entries cancel out in the softmax computation. This means that:*

1. *The optimal denoiser depends only on the pixels that are included in the mask (the support), not their relative weights.*
2. *Binary masks $\{0, 1\}$ are as expressive as any diagonal weighting scheme for this optimization problem.*
3. *This theoretical result justifies our practical choice of binary masks in the main paper, as more complex weighting provides no additional benefit for the optimal denoiser.*

## A.4 "Pass-through" denoisers: detailed analysis of SNR

In this section, we provide a detailed analysis of the signal-to-noise ratio in the principal components of the data, extending section "Pass-through" denoisers in the main paper. Let's consider the data matrix $X = [x_0^1 x_0^2 \ldots x_0^N] \in \mathbb{R}^{d \times N}$. Doing singular value decomposition, and assuming $N \geq d$ we get $X = U diag(\lambda_1, \lambda_2, \ldots \lambda_d) V^T$, where $\lambda_i$ are sorted in the descending order of their absolute values. Covariance of the dataset, assuming that the mean of the dataset is zero:

$$\Sigma = \frac{1}{N} X X^T = \frac{1}{N} U diag(\lambda_1^2, \lambda_2^2, \ldots \lambda_d^2) U^T,$$

where $U$ are the principal components of the data and $\lambda_i^2/N$ is the variance of the data along those components. We can now compute the signal-to-noise ratio along each of the principal components of the data:

$$\text{SNR}_i = \frac{\mathbb{E}_{x_0 \sim X}\left[\left(U_i^T \sqrt{\alpha_t} x_0\right)^2\right]}{\mathbb{E}_{\epsilon \sim \mathcal{N}(0,I)}\left[\left(U_i^T \sqrt{1-\alpha_t}\epsilon\right)^2\right]}$$

$$= \frac{\alpha_t \cdot U_i^T \Sigma U_i}{(1-\alpha_t) \cdot U_i^T U_i}$$

$$= \frac{\alpha_t \cdot \lambda_i^2 / N}{1-\alpha_t}$$

$$= \frac{\lambda_i^2}{N\sigma_t^2}$$

When $\lambda_i^2 \gg N\sigma_t^2$, the intrinsic data variance is much larger than the relative noise level, and the signal was not "destroyed" by noise. Note, the analysis above does not have to be performed on the entire dataset, but rather on the most relevant set of neighbors to the image that is currently being denoised. In that case, the high SNR projections will be more precise and specific to each particular image as long as SVD is well defined. Due to computation constraints and to keep the analysis simple from now on we will assume that the covariance matrix is computed on the entire dataset.

## A.5 Manipulating the sensitivity field: variance of the perturbation

In this section, we provide the derivation for the variance of the added perturbation $\lambda_W$ in the section "Manipulating the sensitivity field" of the main paper. Denote by $v = \gamma c s$ the signal vector; then the empirical covariance of the modified data is

$$\Sigma_{\text{mod}} = \mathbb{E}[\hat{x}_0 \hat{x}_0^T]$$

$$= \mathbb{E}[(x_0 + \gamma c s)(x_0 + \gamma c s)^T]$$

$$= \mathbb{E}[x_0 x_0^T] + \gamma \mathbb{E}[x_0 s^T c^T] + \gamma \mathbb{E}[c s^T x_0^T] + \gamma^2 \mathbb{E}[c s s^T c^T]$$

$$= \Sigma_{\text{orig}} + \gamma^2 \mathbb{E}[c c^T] s s^T$$

$$= \Sigma_{\text{orig}} + \gamma^2 \frac{1}{3} I_3 \otimes s s^T,$$

where we used $\mathbb{E}[x_0] = 0$, $\mathbb{E}[c] = 0$, and for $c \sim \text{Uniform}([-1, 1]^3)$, we have $\mathbb{E}[c c^T] = \frac{1}{3} I$. For the noisy observations $\hat{x}_t = \sqrt{\alpha_t}\hat{x}_0 + \sqrt{1-\alpha_t}\epsilon$, the covariance becomes:

$$\Sigma_{\text{mod}}^t = \mathbb{E}[\hat{x}_t \hat{x}_t^T]$$

$$= \alpha_t \mathbb{E}[\hat{x}_0 \hat{x}_0^T] + (1-\alpha_t)I$$

$$= \alpha_t \Sigma_{\text{mod}} + (1-\alpha_t)I$$

$$= \alpha_t \Sigma_{\text{orig}} + \alpha_t \gamma^2 \frac{1}{3} I_3 \otimes s s^T + (1-\alpha_t)I.$$

Assuming the RGB perturbation affects each color channel independently and focusing on a single channel, the second term contributes a rank-1 perturbation with eigenvalue $\lambda_W^2 = \alpha_t \gamma^2 \|s\|^2 / 3$.

By the Wiener filter analysis of Section 3, the learned sensitivity along the new "W" principal component is

$$s_w(t) = \frac{\lambda_W^2}{\lambda_W^2 + (1-\alpha_t)}$$

$$= \frac{\alpha_t \gamma^2 \|s\|^2 / 3}{\alpha_t \gamma^2 \|s\|^2 / 3 + (1-\alpha_t)}$$

$$= \frac{\alpha_t \gamma^2 \|s\|^2}{\alpha_t \gamma^2 \|s\|^2 + 3(1-\alpha_t)}$$

$$= \frac{\gamma^2 \|s\|^2}{\gamma^2 \|s\|^2 + 3\sigma^2},$$

where $(1-\alpha_t)$ is the noise variance at timestep $t$.

# B  Additional Experiments and Ablation

## B.1  Ablation of our model

The analytical model proposed in this paper has a single hyperparameter: $\tau$ – the threshold of the sensitivity field binarization. In Appendix A.3 we formally justify binarization of the sensitivity fields for our analytical model. Here, we demonstrate the effect of choosing different binarization thresholds. In particular, from Figure 6 we can see that higher threshold values (i.e., smaller patch sizes) correspond to a sharper, but "patchier". On the other side, small threshold values (i.e., bigger patch sizes) cause the generated image to be over-smoothed. We report the $r^2$ and MSE metrics of correlation with the trained diffusion model for different threshold values in Table 2.

Table 2: Comparison of $r^2$ and MSE metrics across datasets for different binarization threshold values. Best values are highlighted in bold.

| Threshold | CIFAR10 | | CelebA-HQ | | MNIST | | Fashion MNIST | |
|---|---|---|---|---|---|---|---|---|
| | $r^2 \uparrow$ | MSE$\downarrow$ | $r^2 \uparrow$ | MSE$\downarrow$ | $r^2 \uparrow$ | MSE$\downarrow$ | $r^2 \uparrow$ | MSE$\downarrow$ |
| *0.005* | 0.396 | 0.059 | 0.786 | 0.038 | **0.492** | **0.151** | **0.563** | **0.115** |
| *0.010* | 0.520 | 0.046 | 0.865 | 0.023 | 0.441 | 0.165 | 0.517 | 0.122 |
| *0.020* | 0.672 | 0.031 | **0.897** | **0.017** | 0.418 | 0.176 | 0.406 | 0.144 |
| *0.050* | **0.773** | **0.021** | 0.894 | 0.017 | 0.214 | 0.255 | 0.072 | 0.211 |
| *0.070* | 0.771 | 0.022 | 0.879 | 0.020 | 0.214 | 0.255 | -0.192 | 0.264 |
| *0.100* | 0.737 | 0.026 | 0.852 | 0.024 | 0.214 | 0.255 | -0.407 | 0.311 |
| *0.150* | 0.641 | 0.036 | 0.799 | 0.033 | 0.214 | 0.255 | -0.209 | 0.270 |

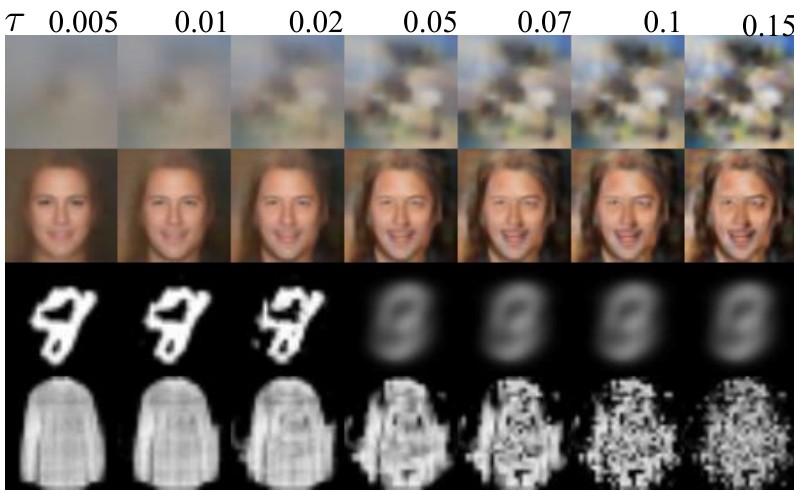

Figure 6: Ablation of the binarization threshold $\tau$.

## B.2  Self-attention layers in denoising U-Nets

Across our experiments, we are using a trained DDPM model with removed self-attention (SA) layers following [12]. In this section, we demonstrate that removing the self-attention layer brings the FID score to $6.04$ from $4.12$ with SA. Qualitatively, the generated images look similar with and without SA, and thus our analysis in the main paper can be extended to U-Nets with SA layers.

In particular, we train a U-Net without self-attention and compare it with a baseline U-Net trained with self-attention. Using the gradient-estimation sampler from [22], we report the FID scores for both models, and in Figure 7, we compare sample results.

DDPM w/o SA (FID 6.04)    DDPM with SA (FID 4.12)

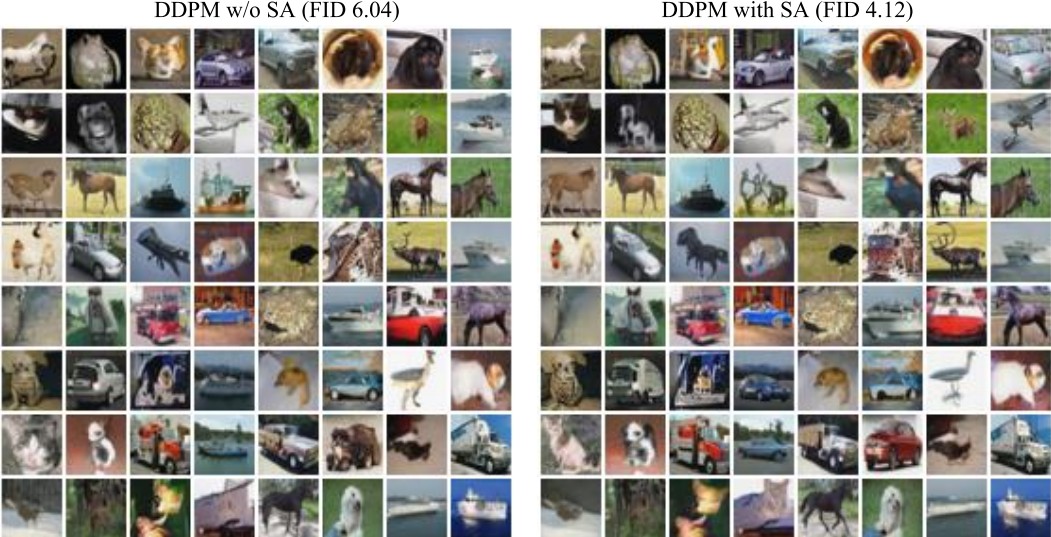

Figure 7: Samples from trained DDPM U-Nets without (left) and with (right) self-attention layers. The initial random noise is the same for both sets of images.

## B.3  Low rank projection of the covariance matrix

In this section, we study whether projecting the covariance matrix to a lower rank can help eliminate the high-frequency artifacts observed in the Wiener Filter in Figure 5.

First, on Figure 8, we analyze the singular values of the covariance matrices for each of the datasets. We observe that most of the total energy is contained in the first 200 singular values for all datasets, leaving a long tail of low-energy principal components.

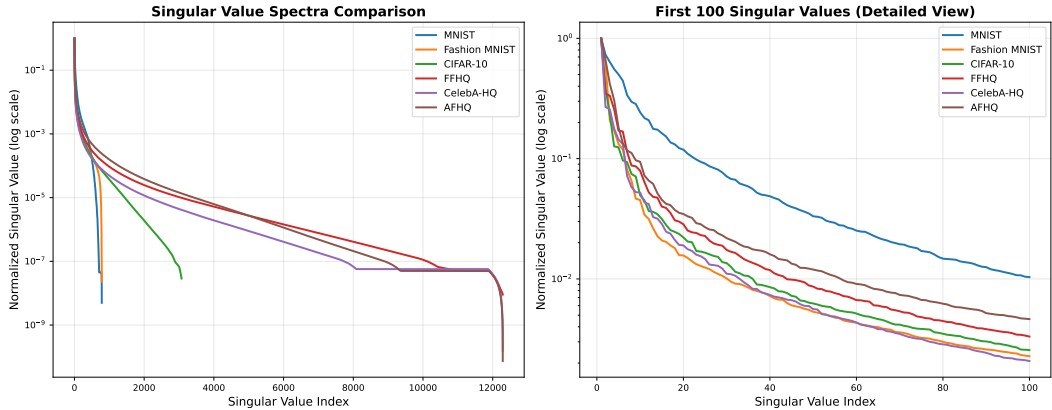

Figure 8: Analysis of the absolute values of the singular values in the covariance matrices across different datasets. Left: across the full set of the singular values. Right: across the first 100 singular values.

Next, we zero out the smallest singular values of the covariance matrix based on different thresholds of total energy. Both for Ours and for the Wiener filter, we measure the difference between predictions of the analytic models and a trained DDPM model, as done in the paper. The results are averaged across 16 generations and presented in Table 3 below. Qualitative results are presented in Figure 9. A 0% SVD Energy Cutoff corresponds to using the full-rank covariance matrix.

For the Wiener filter, low-rank approximation leads to visually smoother outputs; however, it reduces the correlation with outputs from trained diffusion models. Our algorithm does not benefit from using a low-rank projection of the covariance matrix and still demonstrates higher correlation than

Table 3: Comparison of our method vs Wiener filter across different SVD energy cut-off thresholds. Results show R² scores (the higher the better) for each dataset and energy cut-off combination. Best R² score for each dataset is shown in bold.

| SVD Energy Cutoff | 0% | 0.01% | 0.05% | 0.10% | 0.50% | 1.00% | 5.00% | 10.00% | 20.00% | 50.00% |
|---|---|---|---|---|---|---|---|---|---|---|
| Ours \| CIFAR10 | **0.773** | 0.501 | 0.312 | 0.245 | 0.176 | 0.176 | 0.192 | 0.194 | 0.200 | -0.248 |
| Ours \| CELEBA HQ | **0.897** | 0.759 | 0.702 | 0.695 | 0.685 | 0.668 | 0.652 | 0.650 | 0.607 | 0.607 |
| Ours \| MNIST | **0.492** | 0.197 | 0.156 | 0.158 | 0.351 | 0.341 | 0.361 | 0.384 | 0.389 | 0.368 |
| Wiener \| CIFAR10 | 0.674 | **0.677** | 0.661 | 0.649 | 0.538 | 0.460 | 0.223 | 0.168 | 0.127 | -0.048 |
| Wiener \| CELEBA HQ | **0.818** | 0.817 | 0.805 | 0.797 | 0.766 | 0.739 | 0.649 | 0.625 | 0.548 | 0.548 |
| Wiener \| MNIST | **0.469** | 0.468 | 0.468 | 0.467 | 0.453 | 0.445 | 0.421 | 0.394 | 0.353 | 0.292 |

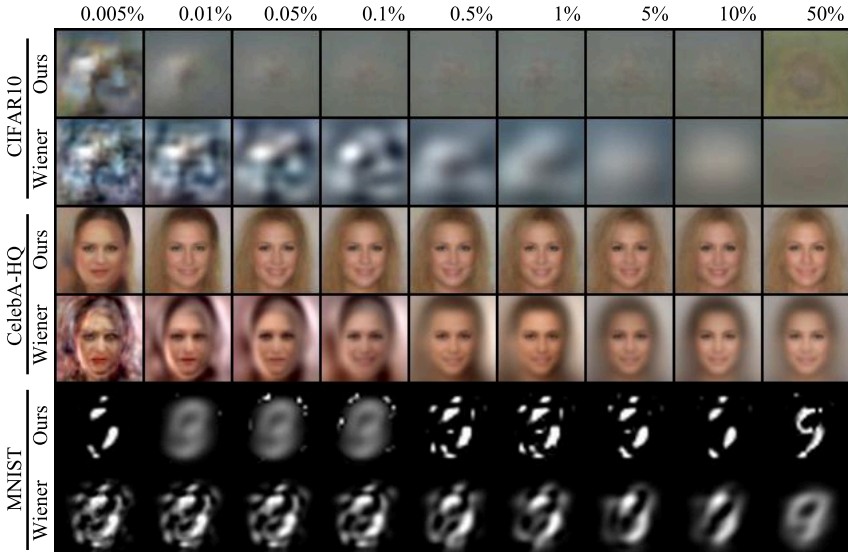

Figure 9: Qualitative comparison between generations of the analytical models when the covariance matrix is projected to a lower rank corresponding to different energy cut-off thresholds.

the Wiener filter. This indicates that low-rank structure alone does not account for the performance of learned denoisers. We believe this result further supports the distinctiveness and relevance of our proposed model.

### B.4 How to reproduce the reported sensitivity fields

In this section, we provide the technical details and intuition needed to measure the sensitivity fields of diffusion models. All the results are reported for CIFAR10 dataset. Recall that the optimization problem is invariant under the change of variables from the initial image $x_0$ to the noise sample $\epsilon$; see Appendix A.1 for details. Consequently, one can measure the sensitivity field in either the noise parameterization, $\partial\epsilon(x,t)/\partial x$, or the image parameterization, $\partial x_0(x,t)/\partial x$. Although the choice is merely a theoretical convenience, in practice the model's behavior is highly sensitive to it.

The *top* row of Figure 10 shows the sensitivity fields of a DDPM model trained to predict $x_0$ and then re-parameterized with a linear transform to predict $\epsilon$; here we plot $\partial\epsilon(x,t)/\partial x$. The *middle* row depicts the same model, but the sensitivity is evaluated in the image parameterization, i.e. $\partial x_0(x,t)/\partial x$. As we can see, a simple linear reparameterization applied to the model output drastically alters the result. These observations are intuitive. From the optimal-denoiser analysis, we know that, in the high-noise regime, the model predicts an image close to the dataset mean. Thus, predicting the added noise sample $\epsilon$ for each pixel $q$ is almost equivalent to outputting $q$ minus that mean, so the noise-parameterized sensitivity field appears highly local. Because this visualization is not very informative, we chose to plot $\partial x_0(x,t)/\partial x$ throughout the paper, as it captures the actual structure of the sensitivity field.

Figure 11 illustrates the effect of training U-Net and DiT models in the two parameterizations. Recall that $x_t = \sqrt{\alpha_t}\, x_0 + \sqrt{1 - \alpha_t}\, \epsilon$. For large $t$ where $\alpha_t \to 0$, image $x_0$ is ill-defined given $\epsilon$ and $x_t$; conversely, for small $t$ where $\alpha_t \to 1$, $\epsilon$ is ill-defined given $x_0$ and $x_t$. Hence, while theory predicts identical results (up to re-parameterization), numerical errors lead to different behavior at low and high noise levels. The top two rows of Figure 11 show $\partial x_0(x,t)/\partial x$ for models trained in the noise parameterization, revealing a pronounced shrinkage of the sensitivity fields in the high-noise regime. We hypothesize that this is a numerical artifact and therefore plot, in the bottom two rows, the fields obtained from models trained directly in the image parameterization. For clarity, all DDPM examples in the main paper are trained in that setting.

Finally, the *middle* and *bottom* rows of Figure 10 compare two normalization strategies. In the middle row, each sensitivity field is normalized independently to $[-1, 1]$; in the bottom row, the images are normalized jointly, preserving relative scale. Joint normalization makes the field appear less local while preserving its overall mass. Throughout the paper, we adopt per-image normalization, as it more faithfully reflects the binarization assumed in our analytical model.

**Summary of visualization choices**

- Train the model to predict the image $x_0$ (not the noise $\epsilon$).
- Visualize the sensitivity of the image prediction, i.e. $\partial x_0(x,t)/\partial x$.
- Apply per-image normalization.

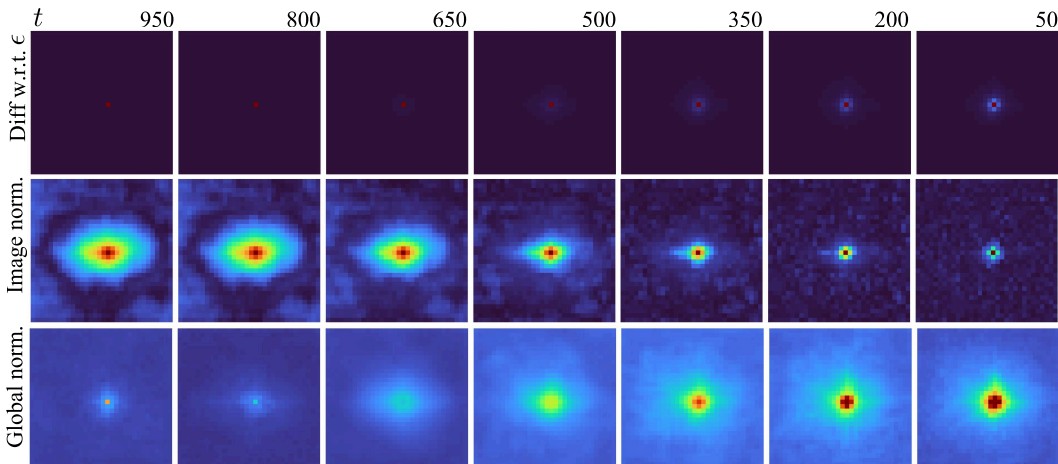

Figure 10: **Top:** sensitivity field of the noise prediction $\partial \epsilon(x,t)/\partial x$. **Middle:** sensitivity field of the image prediction $\partial x_0(x,t)/\partial x$ with per-image normalization. **Bottom:** the same field with joint normalization across images.

### B.5 Sensitivity field of the optimal denoiser

In this section, we provide a visualization of the sensitivity fields of the optimal denoiser on the CIFAR10 dataset. As shown in Figure 12, the sensitivity of the optimal denoiser closely resembles that of the trained models only in the high-noise regime. At intermediate noise levels, the sensitivity field begins to diverge, and in the low-noise regime, it ultimately "explodes".

### B.6 Generation dynamics

Here we provide additional results demonstrating the dynamics of image generation. In Figure 14 we numerically compare $x_0$ predictions through the generation process.

In Figure 13 we demonstrate how the dynamics of image generation of our analytical model compares with that one of a trained DDPM model. Note that the trained model produces noisy single-step predictions for high noise levels ($t \geq 850$). We explain this behavior by the fact that the model was trained to predict $\epsilon$ and later re-parametrized to output $x_0$ for the visualization. Since $\alpha_t \to 0$ for high noise level, $x_0$ becomes ill-defined and thus noises the outputs.

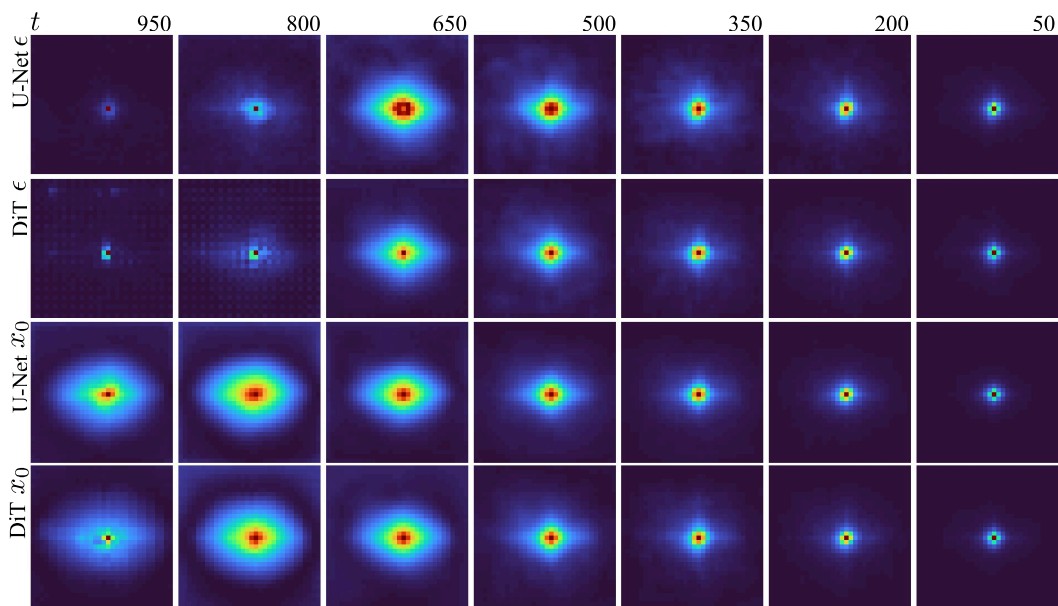

Figure 11: Sensitivity fields $\partial x_0(x,t)/\partial x$ for U-Net (left) and DiT (right). **Top two rows:** models trained to predict noise $\epsilon$. **Bottom two rows:** models trained to predict the image $x_0$. The shrinkage observed at high noise in the noise-parameterized models is likely due to numerical instability.

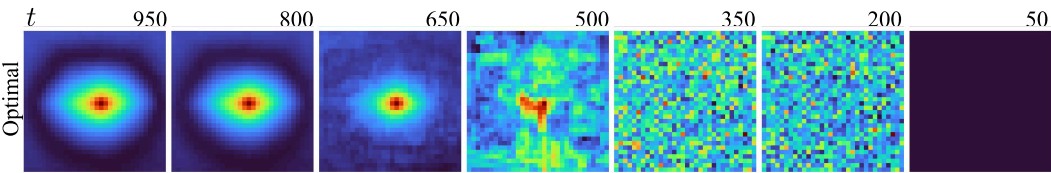

Figure 12: Sensitivity field of the optimal denoiser.

## B.7    Quantitative results for AFHQv2 and Fashion-MNIST

In addition to Table 1 in the main paper we report quantitative results for AFHQv2 and Fashion-MNIST in Table 4. All the values are calculated across 128 samples.

Table 4: Comparison of methods across AFHQv2 and Fashion-MNIST. All metrics are averaged over 128 samples. Best results are highlighted in green and second best in maroon.

|  | AFHQv2 | | Fashion-MNIST | |
|---|---|---|---|---|
| Method | $r^2 \uparrow$ | MSE$\downarrow$ | $r^2 \uparrow$ | MSE$\downarrow$ |
| Optimal Denoiser | -1.239 ± 0.371 | 0.180 ± 0.023 | -0.137 ± 0.344 | 0.254 ± 0.077 |
| Wiener (linear) | 0.601 ± 0.072 | 0.025 ± 0.003 | 0.449 ± 0.068 | 0.137 ± 0.018 |
| Kamb & Ganguli [12] | 0.429 ± 0.081 | 0.041 ± 0.006 | 0.342 ± 0.183 | 0.186 ± 0.019 |
| **Ours** | 0.759 ± 0.026 | 0.019 ± 0.004 | 0.523 ± 0.042 | 0.125 ± 0.011 |
| Another DDPM | 0.928 ± 0.019 | 0.050 ± 0.001 | 0.950 ± 0.020 | 0.015 ± 0.005 |

## B.8    Quantitative measure of novelty of samples

In this work, we focus on the ability of the trained diffusion models to generate novel samples that contrast with the behavior of the optimal denoiser. Therefore, the ability of the analytical model to generate novel samples is paramount. In figure 5 of the main paper (as well as in Appendix B.9) we report the nearest neighbors from the training dataset for each sample generated with our analytical model. To quantify these results, we report the average $L2$ distances between samples generated with

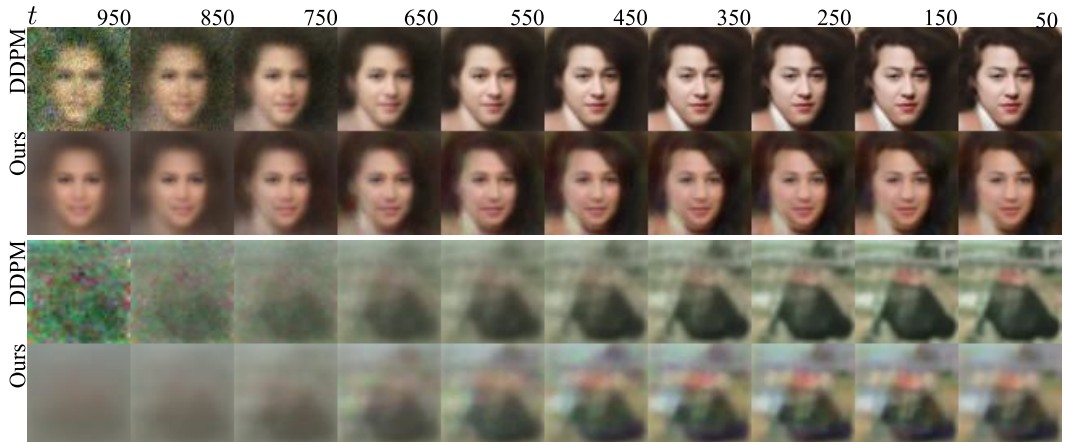

Figure 13: Intermediate generation results of a trained DDPM model (rows 1 and 3) and ours (rows 2 and 4). The figure displays single-step estimations of $x_0$ from each $x_t$ along a sampling trajectory of 10 steps.

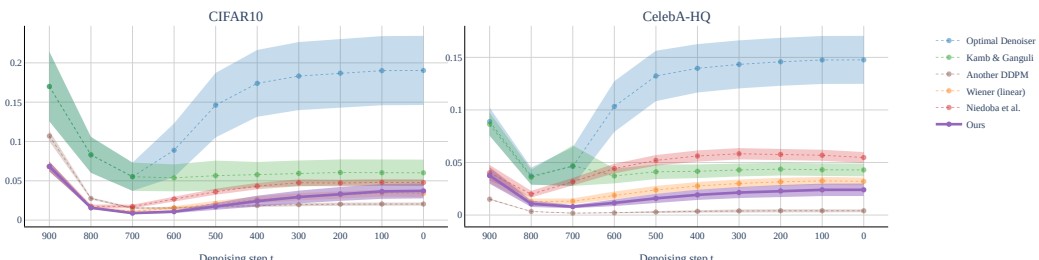

Figure 14: Mean Squared Error (MSE) between the baseline's predictions and a trained DDPM model. The MSE is calculated on $x_0$ prediction from each $x_t$ point along a 10-step generation trajectory. The results are presented on the CIFAR10 and CelebA-HQ datasets. Mean and standard deviation values were calculated across 128 samples.

each of the baseline models and the closest image in the dataset in Table 5. Additionally, we report the dynamics of the "novelty" measure in the generation process in Figure 15.

Table 5: We numerically quantify the ability of analytical models to produce images outside of the training dataset. In this table, we provide the average $L2$ distance between images generated with the baselines and the corresponding closest image in the training dataset. Results are averaged over 128 samples.

| Method | CIFAR10 | CelebA-HQ | AFHQv2 | MNIST | Fashion MNIST |
|---|---|---|---|---|---|
| Optimal Denoiser | $0.000 \pm 0.000$ | $0.000 \pm 0.000$ | $0.000 \pm 0.000$ | $0.000 \pm 0.000$ | $0.000 \pm 0.000$ |
| Wiener (linear) | $0.091 \pm 0.006$ | $0.104 \pm 0.006$ | $0.112 \pm 0.005$ | $0.177 \pm 0.015$ | $0.133 \pm 0.013$ |
| Kamb & Ganguli [12] | $0.094 \pm 0.005$ | $0.089 \pm 0.006$ | $0.136 \pm 0.007$ | $0.355 \pm 0.061$ | $0.218 \pm 0.032$ |
| **Ours** | $0.040 \pm 0.005$ | $0.063 \pm 0.004$ | $0.063 \pm 0.004$ | $0.204 \pm 0.023$ | $0.131 \pm 0.027$ |
| Another DDPM | $0.079 \pm 0.014$ | $0.087 \pm 0.010$ | $0.095 \pm 0.013$ | $0.103 \pm 0.023$ | $0.067 \pm 0.015$ |

## B.9 Additional generation results

We present additional generation results similar to fig. 5 of the main paper in Figures 16 to 20.

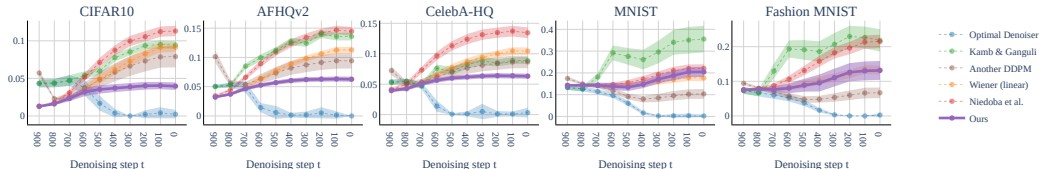

Figure 15: $L2$ distance between $x_0$ prediction and the closest image in the training dataset reported along a 10-step generation trajectory for 5 datasets.

# C Implementation details

## C.1 Sampling.

In all of the generations in this paper, we are using diffusers' [9] implementation of the DDIM [27] sampler with 10 sampling steps. We discretize the noise time scale for $t \in [0, 1000]$ where $t = 0$ is no noise and $t = 1000$ is full noise. The scheduler is linear with $\alpha_0 = 10^{-4}$ and $\alpha_{1000} = 0.02$.

## C.2 Training DDPM Model

We train a Denoising Diffusion Probabilistic Model (DDPM) U-Net using a third-party pytorch implementation [35]. We adopt the U-Net model architecture based on the input image resolution:

- *MNIST/FashionMNIST* (`img_size = 28`): 3 downsampling levels with `channel_mult = [1, 2, 2]`, base channel width 64.
- *CIFAR10* (`img_size = 32`) and *CelebA-HQ/AFHQ* (`img_size = 64`): 4 downsampling levels with `channel_mult = [1, 2, 3, 4]`, base channel width 128.

The number of residual blocks per level is fixed to 2, with no self-attention modules included. Dropout is set to 0.15 throughout the network. The model is trained for 200 epochs with a batch size of 32. We use the Adam optimizer with a learning rate of $10^{-4}$ over 1000 diffusion steps. Training and evaluation use fixed random seeds for reproducibility.

## C.3 Our analytical model

Below, we provide a detailed description of the implementation of our analytical model. A key component of this implementation is the weighted streaming softmax (*wssm*) that accumulates the product $x_0 \operatorname{softmax}(\ldots)$ over batches of training images.

**Algorithm 1** Single denoising step of the proposed analytical model.

---

**Require:** Noisy image $x_t$
    Timestep $t$
    Precomputed covariance $S$ of the data
    Masking threshold $\tau$
    Dataset $X$
    Schedule of $\alpha_t$ and $\sigma_t^2 = \frac{1-\alpha_t}{\alpha_t}$

**Ensure:** Estimated clean image $\hat{x}_0$
1: $U\Lambda U^\top = S$                                              ▷ SVD of the covariance matrix
2: $W_t = \frac{1}{\sqrt{\alpha_t}} U \text{diag}\left(\frac{\lambda_i^2}{\lambda_i^2 + \sigma_t^2}\right) U^\top$        ▷ Current Wiener matrix
3: $M_t = \text{Binarize}(W_t, \tau)$                                  ▷ Construct the projection matrix
4: $wssm.init()$                                                       ▷ Initialize weighted streaming softmax
5: **for** each batch $x_0^{(k)}$ from $X$ **do**
6:     $D_k = stack\left[\left(x_t - \sqrt{\alpha_t}x_0^{(k)}\right)^2\right]$      ▷ Distance to $x_t$ for each image in the batch
7:     $Dm_t = D_k M_t$                                              ▷ Each row of $M_t$ serves as a mask
8:     $wssm.update\left(-Dm_t/2\left[1-\alpha_t\right],\ x_0^{(k)}\right)$       ▷ Add the distances and the value
9: **end for**
10: $\hat{x}_0 = wssm.value()$
11: **return** $\hat{x}_0$

---

## C.4    Baseline implementation details

**Wiener filter.**    To implement the Wiener matrix, we first center each dataset to a zero mean. Then we pre-compute the covariance matrix of the dataset. Note that this is part of "training" and these computations were not included in the runtime report. On sampling, use the PyTorch implementation of SVD to compute the principal components and the corresponding singular values. Finally, we are using eq. (7) from the main paper to implement $W_t$. Note that we are using the Wiener filter as a denoiser, and when generating the images, we are still using a 10-step DDIM sampling, effectively applying the Wiener filter 10 times to the initial noise.

**Kamb & Ganguli model.**    We implemented the analytical model suggested by Kamb & Ganguli in our code base. Then we fit the patch sizes $M_t$ of the analytical model to our trained DDPM U-Nets, maximizing the $r^2$ between the scores on each step of generation. For the CelebA-HQ dataset we are using (non-equivariant) Local Score (LS) machine, for other datasets we used Equivariant Local Score (ELS) machine. Below are the patch sizes that we obtained:

- **CIFAR10** $32 \times 32$: [32, 32, 32, 29, 25, 17, 13, 9, 7, 3]
- **CelebA-HQ** $64 \times 64$: [ 64, 64, 45, 25, 17, 17, 9, 7, 5, 3 ]
- **AFHQ** $64 \times 64$: [64, 64, 45, 33, 25, 17, 17, 9, 9, 3]
- **MNIST** $28 \times 28$: [28, 28, 23, 17, 13, 13, 13, 9, 9, 9]
- **Fashion MNIST** $28 \times 28$ : [28, 25, 23, 17, 17, 13, 13, 9, 9, 5]

## C.5    Algorithmic Complexity

In this section, we provide an analysis of the algorithmic complexity of our method and the baselines. For small-resolution images, the Wiener filter remains the most efficient at $O(m^2)$, where $m$ is the flattened image resolution. Both our model and Kamb&Ganguli's require a dataset pass per inference step, leading to scaling linear in $n$, where $n$ is the dataset size.

Kamb&Ganguli assume translation equivariance, so for each pixel, its surrounding patch (size $p_t$ at denoising step $t$) is compared to every patch in the dataset, resulting in $O(np_t m^2)$ complexity. With approximate vector search (e.g., using $k$ clusters), this reduces to $O(np_t m^2/k)$. Our model forgoes translation equivariance as we did not observe any difference in quality. Additionally, we are using a distinct per-pixel mask pattern. This leads to the algorithmic complexity of our algorithm being

$O(np_tm)$. For larger datasets, we can match the $O(np_tm/k)$ complexity by indexing masks per timestep and pixel. We provide the summary in Appendix C.5.

Table 6: Algorithmic complexity of the baselines. Here, $m$ denotes the flattened image resolution, $n$ the dataset size, $p_t$ the patch size at denoising step $t$, and $k$ the number of clusters used in approximate nearest-neighbor search.

| Method | Wiener | Kamb (exact) | Kamb (approx) | Ours (exact) | Ours (approx) |
|---|---|---|---|---|---|
| Complexity | $O(m^2)$ | $O(np_tm^2)$ | $O\left(\frac{np_tm^2}{k}\right)$ | $O(np_tm)$ | $O\left(\frac{np_tm}{k}\right)$ |

### C.6 Computational resources and runtime

All the experiments were performed on a server machine with *Ubuntu 20.04*. The machine has *1008GB* RAM, *128* CPU cores and $8\times$ *NVIDIA RTX A6000* GPUs with *49140MB* VRAM. We note that all the baselines could be run with fewer computational resources. In Table 7 we provide the average run times for each baseline.

Table 7: We demonstrate the computational efficiency of each method by displaying the total sampling time for each of the baselines over 10 denoising steps. None of the methods are optimized for runtime, and the comparison is provided only as a rough reference. Results show times averaged over 64 samples.

| Method | CIFAR10 | CelebA-HQ | AFHQv2 | MNIST | Fashion MNIST |
|---|---|---|---|---|---|
| Optimal Denoiser | 7.90 | 18.90 | 10.01 | 0.63 | 0.64 |
| Wiener (linear) | 0.11 | 3.10 | 3.08 | 0.07 | 0.07 |
| Kamb & Ganguli [12] | 44.44 | 349.68 | 181.08 | 4.40 | 4.49 |
| **Ours** | 21.25 | 70.23 | 314.55 | 22.39 | 22.97 |
| Another DDPM | 0.57 | 0.65 | 0.65 | 0.61 | 0.63 |

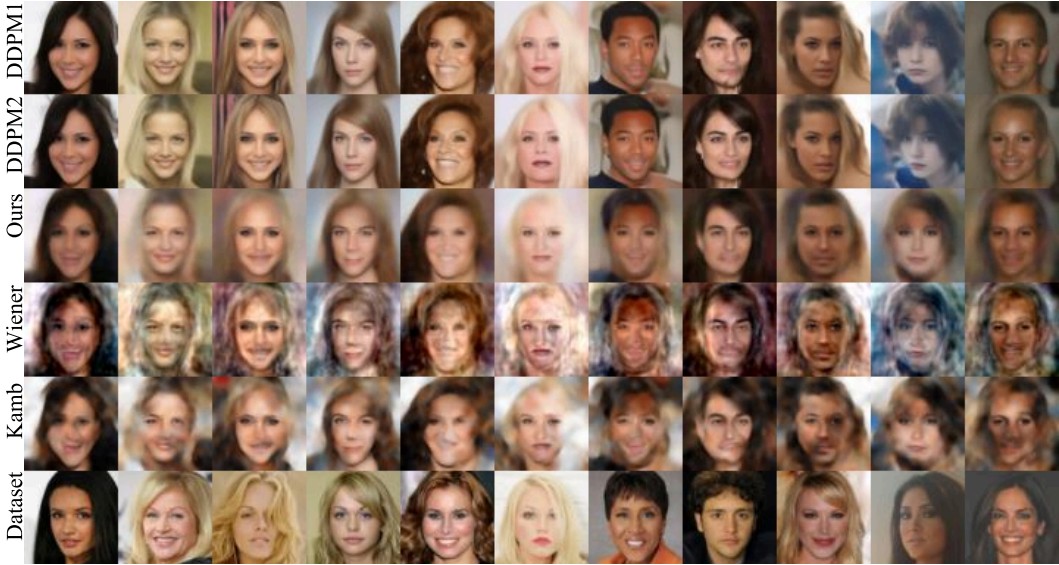

Figure 16: Additional generation results for all baselines and ours on the CelebA-HQ dataset.

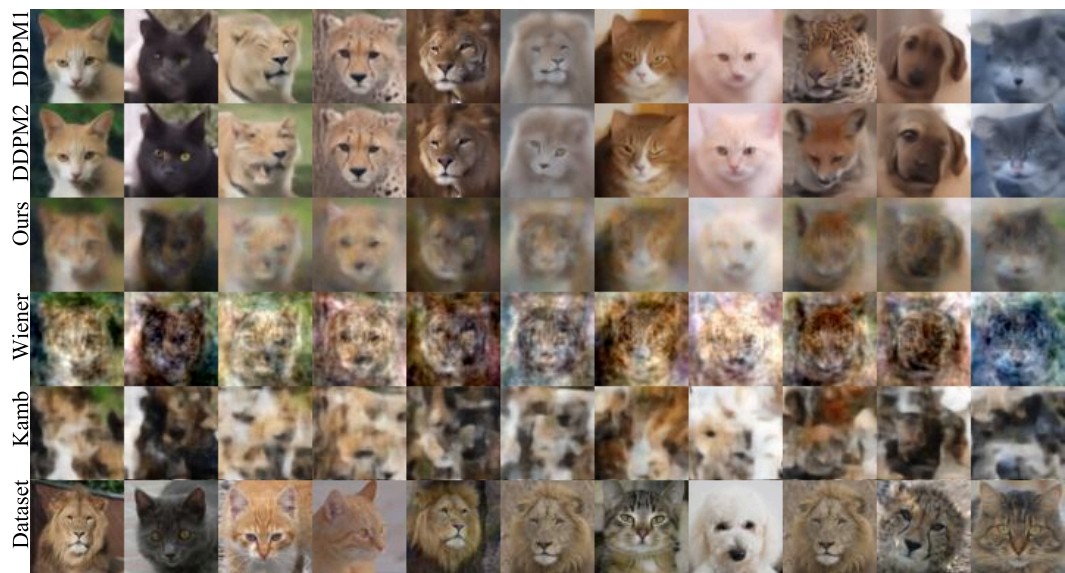

Figure 17: Additional generation results for all baselines and ours on the AFHQ dataset.

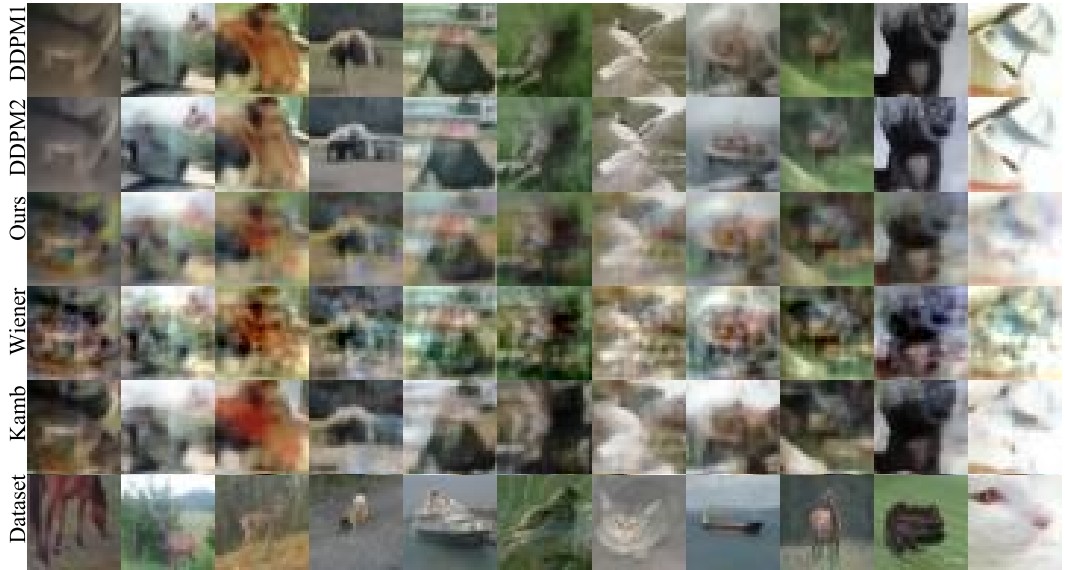

Figure 18: Additional generation results for all baselines and ours on the CIFAR10 dataset.

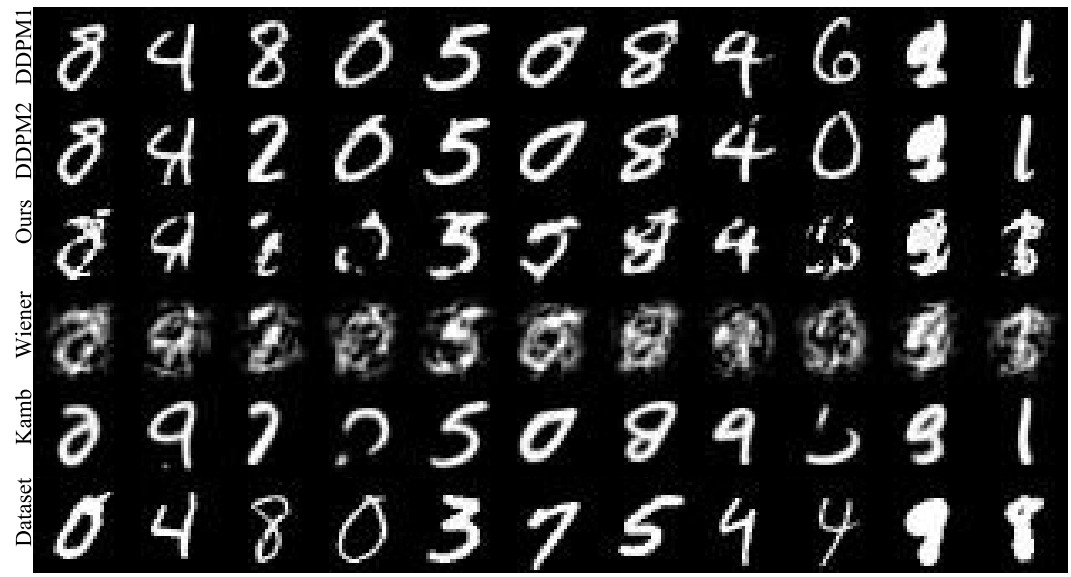

Figure 19: Additional generation results for all baselines and ours on the MNIST dataset.

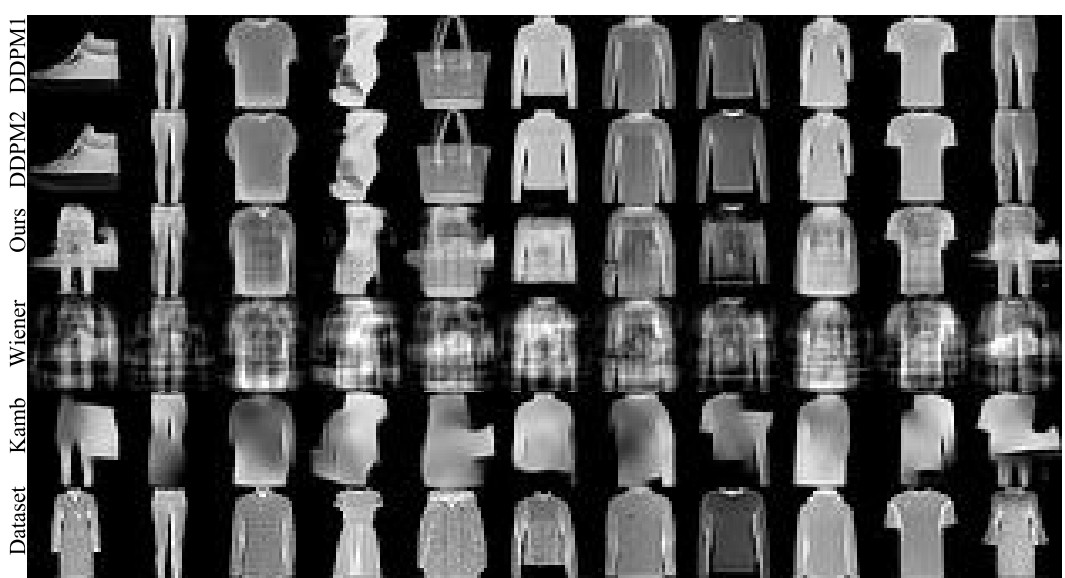

Figure 20: Additional generation results for all baselines and ours on the Fashion MNIST dataset.

