# OpenReview forum: "Locality in Image Diffusion Models Emerges from Data Statistics"
_NeurIPS.cc/2025/Conference — NeurIPS 2025 spotlight_

### Official Review · Reviewer_Ws6m · 2025-06-19

**Clarity:** 3
**Significance:** 3
**Originality:** 3
**Rating:** 5
**Confidence:** 5

**Summary:**

This work demonstrates that the locality in image diffusion models arises from the structure of data statistics (covariance structure) rather than the architectural inductive bias of the deep networks. They support this claim by showing  linear diffusion models consisting of Wiener filters which are purely determined by the data statistics outperform previously proposed analytical diffusion models based on architectural inductive bias. Building upon this finding, they replace the sensitivity fields used in previous patch-based analytical diffusion models with the binarized version of the sensitivity files of the linear diffusion model, which leads to improved performance.

**Questions:**

1. What are other choices of $A_t^q$? What do you think is the limit of the patch based analytical denoiser, suppose the optimal  $A_t^q$ can be found? What would be the property of the optimal $A_t^q$?

2. Can you do ablation study on the effects of first doing low-rank approximation on the covariance matrix and then build the Wiener filter or the proposed method? Can low-rank approximation further improve the performance?

**Ethical Concerns:**

["NO or VERY MINOR ethics concerns only"]

**Final Justification:**

The rebuttal provides extra experiments and have addressed most of my questions. I believe the paper has make progress towards understanding the mechanism of generalization of diffusion models.

**Limitations:**

See the "Strengths and Weakness" section.

**Paper Formatting Concerns:**

No concerns.

**Quality:**

3

**Strengths And Weaknesses:**

Strengths:

1. This finding—that locality in image diffusion models derives from the data’s covariance structure rather than from the networks’ architectural inductive biases—has important implications, correcting the common belief that the models’ creative power primarily originates from network architecture [1].

2. The proposed method improves upon previously proposed analytical diffusion models, hence contributing to the understanding of the generalization mechanisms of diffusion models.

Weaknesses:

1. The proposed model in equation (9) is kind of abrupt. While I understand this choice exploit the second order statistics of the dataset and also integrate the binary threshold as inspired by proposition A.10, it is still simple and fail short at  capturing more complex data structures. A natural question to ask is what is the limitation of tuning $A_t^q$? Can you make objective (8) in A.3 differentiable and tuning the sensitivity field end to end? I understand the Mahalanobis distance makes implementation difficult but I am curious if we can make some approximations.

2. From my experience, building the linear denoisers with low-rank approximation of the full covariance matrices can bring significant improvement, visually similar to the results of (ours) presented in Figure 5. I encourage the author to document the performance of linear denoisers under different low-rank approximations. Similarly, the proposed method might also benefit from low rank approximation. Intuitively, many tail principal components can correspond to noise rather than informative images structures. Get rid of those can potentially improve the denoising performance.

3. Section A, especially the part you motivate the proposed method should be put in the main text.

4. There is a typo in line 171 of the appendix.

Kamb, Mason, and Surya Ganguli. "An analytic theory of creativity in convolutional diffusion models." arXiv preprint arXiv:2412.20292 (2024).

---

> ### Author Rebuttal · Authors · 2025-07-31
>
> We thank Reviewer Ws6m for recognizing the broader implications of our findings, especially in “correcting the common belief” about generalization in diffusion models and for noting that our method “improves upon previously proposed analytical diffusion models.” Other reviewers, such as 8EDr and 8kUK, also appreciated our benchmarks, experimental rigor, and the clarity of our comparative analysis.
>
> We also appreciate reviewers' suggestions on how to further strengthen the paper. In the rest of this reply, we first address suggestions shared by multiple reviewers and then will specifically address comments in this review.
>
> # Shared comments
>
> ## Computational Efficiency
>
> Reviewers 8EDr and 8kUK suggested including a comparison of runtime between the baselines. The runtimes and hardware used for all baselines are reported in Section C.5 of the appendix, and we provide additional context below. Our algorithm is more computationally efficient than Kamb & Ganguli's model both asymptotically and in practice, as we are not assuming translation equivariance. As reviewer 8kUK correctly mentioned, creating an efficient analytical denoiser is not the main focus of this paper; however, we agree that additional analysis of computational efficiency will further strengthen the paper.
>
> Below, we provide a detailed analysis of the algorithmic complexity for the Wiener filter, Kamb & Ganguli's analytical model, and ours.
>
> ### Algorithmic Complexity and the Softmax
>
> For small-resolution images, the Wiener filter remains the most efficient at $O(m^2)$, where $m$ is the flattened image resolution. Both our model and Kamb & Ganguli’s require a dataset pass per inference step, leading to scaling linear in $n$, where $n$ is the dataset size.
>
> Kamb & Ganguli assume translation equivariance, so for each pixel, we compare its surrounding patch (size $p_t$ at denoising step $t$) to every patch in the dataset, resulting in $O(n p_t m^2)$ complexity. With approximate vector search (e.g., using $k$ clusters), this reduces to $O(n p_t m^2 / k)$. Our model forgoes translation equivariance as we did not observe any difference in quality (reported in the main paper). Additionally, we are using a distinct per-pixel mask pattern. This leads to the algorithmic complexity of our algorithm being $O(n p_t m)$. While this makes approximate search harder, our implementation (Section C.5) is faster than even Kamb & Ganguli’s approximate version on benchmark-scale datasets. For larger datasets, we can match their $O(n p_t m / k)$ complexity by indexing masks per timestep and pixel.
>
> **Softmax**: As Reviewer 8kUK noted, computing softmax across a dataset is challenging. We address this with a numerically stable streaming implementation requiring just one pass and constant memory (see `src/utils.py` in the appendix files).
>
> **Algorithmic complexity table**:
> |Method|Wiener Filter|Kamb & Ganguli (exact)|Kamb & Ganguli (approx)|Ours (exact, implemented)|Ours (approx, not implemented)|
> |-|-|-|-|-|-|
> |Complexity|$O(m^2)$|$O(n p_t m^2)$|$O\left(\frac{n p_t m^2}{k}\right)$|$O(n p_t m)$| $O\left(\frac{n p_t m}{k}\right)$|
>
>
> ## Choice of $A_t^q$: intuition, other alternatives and formatting
>
> ### Motivation in Section A.3 (8kUK, Ws6m)
>
> We agree with reviewers 8kUK and Ws6m that the main paper will benefit from some part of the motivation of our algorithm from the appendix to the main paper. We will update the main paper with a clearer intuition behind our algorithm: Kamb and Ganguli use rectangular binary masks fitted to the sensitivity fields of trained denoisers. In our paper, we show that the sensitivity fields of the trained denoisers come from the data covariance. In this paper, we are considering constant sensitivity fields, i.e., $A_t^q$ is not a function of $x$. If the sensitivity field is constant, the denoiser is linear, and we know that the optimal linear denoiser is the Wiener filter. Thus, the sensitivity field under the constant constraint is that of the Wiener filter.
>
> We also revised the derivations for why the sensitivity fields should be binary:
>
> * **Generalized locality definition** We first consider a generalized notion of locality, with the constraint $f^q(x) = f^q(A^q_t x)$, where $A^q_t$ can be any matrix. Please note that indeed, when $A^q_t$ is invertible, $f^q$ is effectively unconstrained, i.e., there is no locality. When $A^q_t = 0$, $f^q$ does not depend on $x$.
>
> * **Generalized locality imposes subspace projection** Next we prove that imposing the generalized locality constraint is equivalent to adding a projection operator $\mathbf{P_t^q}=(A_t^q)^T (A_t^q (A_t^q)^T)^\dagger A_t^q$ (projection onto the row-space of $A^q_t$) to the optimal denoiser: $\hat{f}^q(x, t) = \sum_{i=1}^N (x^i_0)^q \; \text{softmax}_i\left(-\frac{1}{2\sigma_t^2} \|\mathbf{P_t^q}(x - \sqrt{\alpha_t}x_0^i)\|^2\right)$
>
> * **Projection = Orthogonal change of basis + Masking** All projection operators can be written as an orthogonal change of basis $V_t^q$ followed by a masking operation $M^q_t$ (i.e., a diagonal 0-1 matrix): $\mathbf{P_t^q} = (V_t^q)^TM^q_tV_t^q$
>
> This explains why the only interesting generalized locality matrices are binary masks. We will move motivation for our method and additional intuition from Appendix A.3 to Section 3 of the main paper, in case if the paper is accepted. We will update our paper's appendix with proofs of the above statements.
>
> # Individual comments
>
> ## RE: What are other choices of $A_t^q$
> Thank you for your thoughtful comments and suggestions on the choice of $A_t^q$. We provided extended motivation earlier in this reply. We also attempted to perform gradient-based optimization of the matrix $A_t^q$ pre-binarization; however, in the limited time of the rebuttal, we were not able to arrive at conclusive results.
>
> Overall, our intuition is that for constant sensitivity fields (i.e., not a function of $x$), the denoiser is linear, and we know that the optimal linear denoiser is the Wiener filter. To further close the gap between analytical and trained denoisers, we will need to allow the sensitivity fields to depend on the input image, i.e., $A_t^q(x_t)$ should be a function of the noisy image. We agree that this is a very interesting direction to explore and plan to address it in future research.
>
> ## RE: Low-Rank Covariance Matrix
> We agree that a low-rank projection of the covariance matrix is a good method to prevent high-frequency artifacts in the generations produced by the Wiener filter. We performed additional experiments to see if removing the weakest principal components in the covariance matrix can help the analytical models better explain trained diffusers. For the Wiener filter, low-rank approximation leads to visually smoother outputs; however, it reduces the correlation with outputs from trained diffusion models. Our algorithm does not benefit from using a low-rank projection of the covariance matrix and still demonstrates higher correlation than the Wiener filter. This indicates that low-rank structure alone does not account for the performance of learned denoisers. We believe this result further supports the distinctiveness and relevance of our proposed model. Please refer to our detailed qualitative and quantitative analysis below.
>
> ### Singular Values
> First, analyzed the singular values of the covariance matrices for each of the datasets. We observe that most of the total energy is contained in the first 10% of the singular values for all datasets, leaving a long tail of principal components. We plan to include the plots in the appendix of the final version if accepted.
>
> ### Quantitative Comparison
> We zero out the smallest singular values of the covariance matrix based on different thresholds of total energy. Both for ours and for the Wiener filter we measure the $r^2$ between predictions of the analytic models and a trained DDPM model, as done in the paper. The results are averaged across 16 generations and presented in the table below. A 0% SVD Energy Cutoff corresponds to using the full-rank covariance matrix.
>
> |SVD Energy Cutoff|0%|0.01%|0.05%|0.10%|0.50%|1.00%|5.00%|10.00%|50.00%|
> |-|-|-|-|-|-|-|-|-|-|
> |**Ours \| CIFAR10**|**0.588**|0.501|0.312|0.245|0.176|0.176|0.192|0.194|-0.248|
> |**Ours \| CELEBA HQ**|**0.897**|0.759|0.702|0.695|0.685|0.668|0.652|0.650|0.607|
> |**Ours \| MNIST**|**0.492**|0.197|0.156|0.158|0.351|0.341|0.361|0.384|0.368|
> |**Wiener \| CIFAR10**|**0.410**|0.408|0.409|0.382|0.338|0.260|0.223|0.168|-0.048|
> |**Wiener \| CELEBA HQ**|**0.818**|0.817|0.805|0.797|0.766|0.739|0.649|0.625|0.548|
> |**Wiener \| MNIST**|**0.469**|0.468|0.468|0.467|0.453|0.445|0.421|0.394|0.292|
>
> ### Qualitative Comparison
> Additionally, we generate visual comparisons between all the baselines across the different energy cut-off thresholds. We observe that removing a small portion of the singular values (e.g., 0.01% in total energy) indeed smooths out the images generated by the Wiener filter and makes the results more visually appealing. However, as we can see from the table above, this does not lead to a better correlation with trained diffusers. For our method, projecting the covariance matrix to a low rank does not seem to visually improve the results—it only blurs the images. We plan to include the visual comparisons in the final version of the paper if accepted. *Please note that we cannot attach the images to this response due to the new NeurIPS guidelines*.
>
> ### Conclusion
> *In conclusion, based on both qualitative and quantitative comparisons across multiple energy cut-off thresholds, we did not find evidence that low-rank projection of the covariance matrix in analytical models leads to a better explanation of trained denoisers.*
>
> ### RE: typo
> Thank you for highlighting the typo. We fixed it.
>
> Thank you for your thoughtful review, and please let us know if you have any additional suggestions on how we can further improve our paper.

---

### Official Review · Reviewer_8EDr · 2025-06-30

**Clarity:** 3
**Significance:** 3
**Originality:** 3
**Rating:** 4
**Confidence:** 2

**Summary:**

This study challenges the prevailing assumption that CNN inductive biases primarily explain the gap between optimal diffusion denoisers and practical applications, and demonstrates that locality arises from underlying statistics of image datasets. The authors develop an analytical denoiser based on principal component analysis based on these statistical principles, which yields results that are closer to the true diffusion model behaviour than previous expert-designed approximations. It shows that local sensitivity in diffusion models arises from learned data statistics rather than architectural inductive biases, and identifies that analytically derived optimal linear filters-dependent only on training data, accurately predict both local and non-local spatial sensitivities observed in trained denoising networks, unexpectedly outperforming previous analytical models based on optimal denoisers on new quantitative benchmarks.

**Questions:**

Refer to weaknesses part.

**Ethical Concerns:**

["NO or VERY MINOR ethics concerns only"]

**Final Justification:**

The authors' ​​response​​ has addressed my concerns, so ​​I will maintain my rating.​​

**Limitations:**

Yes

**Quality:**

3

**Strengths And Weaknesses:**

This work demonstrates that local sensitivity in diffusion models arises from learned data statistics rather than architectural inductive biases. It is demonstrated that analytically derived optimal linear filters which depend only on training data can accurately predict local and non-local spatial sensitivities observed in trained denoising networks, and surprisingly outperform previous analytical models based on optimal denoisers on new quantitative benchmarks. ​​ ​​The authors explore the mechanism of locality in image diffusion models from the perspective of data statistics and provide detailed visual proofs.

Weaknesses: The author gives a very detailed discussion about his proposed strategy, but it seems to increase some computational efficiency, so I am curious about the experimental results of this. Although the method performs significantly, a key practical consideration is the computational overhead. Could the authors provide the above results? I am not very familiar with this field; therefore, if there are enough reviewers, please ignore my comments.

---

> ### Author Rebuttal · Authors · 2025-07-31
>
> Thank you for the thorough review of our paper, highlighting that it “surprisingly outperforms previous analytical models,” and for appreciating our “new quantitative benchmarks,” “detailed visual proofs,” and “very detailed discussion.” We're also encouraged that other reviewers, such as 8kUK and 4nQU, similarly praised the clarity, rigor, and completeness of our experiments and comparisons.
>
> We also appreciate reviewers' suggestions on how to further strengthen the paper. In the rest of this reply, we first address suggestions shared by multiple reviewers and then will specifically address comments in this review.
>
> ## Computational Efficiency
>
> Reviewers 8EDr and 8kUK suggested including a comparison of runtime between the baselines. The runtimes and hardware used for all baselines are reported in Section C.5 of the appendix, and we provide additional context below. Our algorithm is more computationally efficient than Kamb & Ganguli's model both asymptotically and in practice, as we are not assuming translation equivariance. As reviewer 8kUK correctly mentioned, creating an efficient analytical denoiser is not the main focus of this paper; however, we agree that additional analysis of computational efficiency will further strengthen the paper.
>
> Below, we provide a detailed analysis of the algorithmic complexity for the Wiener filter, Kamb & Ganguli's analytical model, and ours.
>
> ### Algorithmic Complexity and the Softmax
>
> For small-resolution images, the Wiener filter remains the most efficient at $O(m^2)$, where $m$ is the flattened image resolution. Both our model and Kamb & Ganguli’s require a dataset pass per inference step, leading to scaling linear in $n$, where $n$ is the dataset size.
>
> Kamb & Ganguli assume translation equivariance, so for each pixel, we compare its surrounding patch (size $p_t$ at denoising step $t$) to every patch in the dataset, resulting in $O(n p_t m^2)$ complexity. With approximate vector search (e.g., using $k$ clusters), this reduces to $O(n p_t m^2 / k)$. Our model forgoes translation equivariance as we did not observe any difference in quality (reported in the main paper). Additionally, we are using a distinct per-pixel mask pattern. This leads to the algorithmic complexity of our algorithm being $O(n p_t m)$. While this makes approximate search harder, our implementation (Section C.5) is faster than even Kamb & Ganguli’s approximate version on benchmark-scale datasets. For larger datasets, we can match their $O(n p_t m / k)$ complexity by indexing masks per timestep and pixel.
>
> **Softmax**: As Reviewer 8kUK noted, computing softmax across a dataset is challenging. We address this with a numerically stable streaming implementation requiring just one pass and constant memory (see `src/utils.py` in the appendix files).
>
> **Algorithmic complexity table**:
> |Method|Wiener Filter|Kamb & Ganguli (exact)|Kamb & Ganguli (approx)|Ours (exact, implemented)|Ours (approx, not implemented)|
> |-|-|-|-|-|-|
> |Complexity|$O(m^2)$|$O(n p_t m^2)$|$O\left(\frac{n p_t m^2}{k}\right)$|$O(n p_t m)$| $O\left(\frac{n p_t m}{k}\right)$|
>
>
> ## Choice of $A_t^q$: intuition, other alternatives and formatting
>
> ### Motivation in Section A.3 (8kUK, Ws6m)
>
> We agree with reviewers 8kUK and Ws6m that the main paper will benefit from some part of the motivation of our algorithm from the appendix to the main paper. We will update the main paper with a clearer intuition behind our algorithm: Kamb and Ganguli use rectangular binary masks fitted to the sensitivity fields of trained denoisers. In our paper, we show that the sensitivity fields of the trained denoisers come from the data covariance. In this paper, we are considering constant sensitivity fields, i.e., $A_t^q$ is not a function of $x$. If the sensitivity field is constant, the denoiser is linear, and we know that the optimal linear denoiser is the Wiener filter. Thus, the sensitivity field under the constant constraint is that of the Wiener filter.
>
> We also revised the derivations for why the sensitivity fields should be binary:
>
> * **Generalized locality definition** We first consider a generalized notion of locality, with the constraint $f^q(x) = f^q(A^q_t x)$, where $A^q_t$ can be any matrix. Please note that indeed, when $A^q_t$ is invertible, $f^q$ is effectively unconstrained, i.e., there is no locality. When $A^q_t = 0$, $f^q$ does not depend on $x$.
>
> * **Generalized locality imposes subspace projection** Next we prove that imposing the generalized locality constraint is equivalent to adding a projection operator $\mathbf{P_t^q}=(A_t^q)^T (A_t^q (A_t^q)^T)^\dagger A_t^q$ (projection onto the row-space of $A^q_t$) to the optimal denoiser: $\hat{f}^q(x, t) = \sum_{i=1}^N (x^i_0)^q \; \text{softmax}_i\left(-\frac{1}{2\sigma_t^2} \|\mathbf{P_t^q}(x - \sqrt{\alpha_t}x_0^i)\|^2\right)$
>
> * **Projection = Orthogonal change of basis + Masking** All projection operators can be written as an orthogonal change of basis $V_t^q$ followed by a masking operation $M^q_t$ (i.e., a diagonal 0-1 matrix): $\mathbf{P_t^q} = (V_t^q)^TM^q_tV_t^q$
>
> This explains why the only interesting generalized locality matrices are binary masks. We will move motivation for our method and additional intuition from Appendix A.3 to Section 3 of the main paper, in case if the paper is accepted. We will update our paper's appendix with proofs of the above statements.
>
> ### Low-Rank Covariance Matrix
> In response to Reviewer Ws6m’s suggestion, we conducted an additional comparison between the Wiener filter and our algorithm under low-rank projections of the covariance matrix. For the Wiener filter, low-rank approximation leads to visually smoother outputs; however, it reduces the correlation with outputs from trained diffusion models. Our algorithm does not benefit from using a low-rank projection of the covariance matrix and still demonstrates higher correlation than the Wiener filter. This indicates that low-rank structure alone does not account for the performance of learned denoisers. We believe this result further supports the distinctiveness and relevance of our proposed model. Please refer to our detailed qualitative and quantitative analysis in the personal response to Reviewer Ws6m.
>
> Please let us know if you have any additional suggestions on how we can further improve our paper.

---

### Official Review · Reviewer_4nQU · 2025-06-30

**Clarity:** 4
**Significance:** 3
**Originality:** 3
**Rating:** 5
**Confidence:** 3

**Summary:**

The paper challenges the current understanding of the generalization capabilities of diffusion models. Specifically, the authors analyze the gap between the optimal denoiser and approximate denoisers learnt by neural networks. It is demonstrated, theoretically and empirically, that the locality of the learnt denoisers can be explained by data statistics, not by inductive biases of the neural network architectures.

**Questions:**

1. Is the difference between the proposed method and the baselines in Table 1 statistically significant in all experiments?
2. I think that Figure 5 should be improved. My suggestion would be to remove the last row unless the authors can convince me that it is relevant to that experiment (in which case, I believe that it should also be presented for other baselines)

**Ethical Concerns:**

["NO or VERY MINOR ethics concerns only"]

**Final Justification:**

I maintain my support for the paper. I chose not to give a higher score, because I am not an expert in this area.

**Limitations:**

The limitations are properly discussed.

**Paper Formatting Concerns:**

I did not spot any formatting issues.

**Quality:**

4

**Strengths And Weaknesses:**

**Strengths:**
1. This paper is very well-written. It reads somewhat like a tutorial, which I appreciate. The reader is guided through the problem statement, current problem understanding and approaches, experiment design, and results.
2. I believe that the research problem is important, especially given that this work challenges the understanding of the machine learning community and provides an alternative explanation of the generalization capabilities of denoising diffusion models.
3. The authors provide the code to facilitate the replication of results. The codebase is neatly organized.
4. Experiments are clearly explained and convincingly support the claims made by the authors.
5. Proofs in the appendix are readable and easy to follow, given the explanations throughout.

**Weaknesses:**
I did not spot any major weaknesses. Minor weaknesses/suggestions:

1. The README in the codebase could be written better. There are typos, and not immediately obvious which scripts should be run. This should be improved upon acceptance.
2. Figure 5 - at a first glance, the "Ours NN" label can be interpreted (at least I initially did) as "Ours - Neural Network", as in "We built a Neural Network model using our theoretical insights". Perhaps it can be relabeled somehow to instantly be read as "nearest neighbor in the data"?
    * On second thought, I am not sure I understand what is the purpose of this last row. Why show this only for the proposed method, and not other baselines?
3. Table 1 - would be nice to have some error bars/statistical significance tests. Although the difference sometimes seems quite large.
    * I can see that the standard deviations are reported separately in a table in the appendix, but that is not very readable to me, having to go back and forth looking and the values table and the standard deviation table

---

> ### Author Rebuttal · Authors · 2025-07-31
>
> We sincerely thank Reviewer 4nQU for highlighting the strength of our approach, in particular, evaluating it as “very well-written”, as well as your remarks on the importance of the problem and the clarity of our experiments and proofs. We’re also grateful that other reviewers, including 8EDr and 8kUK, found the benchmarks and comparisons both novel and convincing.
>
> We also appreciate reviewers' suggestions on how to further strengthen the paper. In the rest of this reply, we first address suggestions shared by multiple reviewers and then will specifically address comments in this review.
>
> # Shared comments
>
> ## Computational Efficiency
>
> Reviewers 8EDr and 8kUK suggested including a comparison of runtime between the baselines. The runtimes and hardware used for all baselines are reported in Section C.5 of the appendix, and we provide additional context below. Our algorithm is more computationally efficient than Kamb & Ganguli's model both asymptotically and in practice, as we are not assuming translation equivariance. As reviewer 8kUK correctly mentioned, creating an efficient analytical denoiser is not the main focus of this paper; however, we agree that additional analysis of computational efficiency will further strengthen the paper.
>
> Below, we provide a detailed analysis of the algorithmic complexity for the Wiener filter, Kamb & Ganguli's analytical model, and ours.
>
> ### Algorithmic Complexity and the Softmax
>
> For small-resolution images, the Wiener filter remains the most efficient at $O(m^2)$, where $m$ is the flattened image resolution. Both our model and Kamb & Ganguli’s require a dataset pass per inference step, leading to scaling linear in $n$, where $n$ is the dataset size.
>
> Kamb & Ganguli assume translation equivariance, so for each pixel, we compare its surrounding patch (size $p_t$ at denoising step $t$) to every patch in the dataset, resulting in $O(n p_t m^2)$ complexity. With approximate vector search (e.g., using $k$ clusters), this reduces to $O(n p_t m^2 / k)$. Our model forgoes translation equivariance as we did not observe any difference in quality (reported in the main paper). Additionally, we are using a distinct per-pixel mask pattern. This leads to the algorithmic complexity of our algorithm being $O(n p_t m)$. While this makes approximate search harder, our implementation (Section C.5) is faster than even Kamb & Ganguli’s approximate version on benchmark-scale datasets. For larger datasets, we can match their $O(n p_t m / k)$ complexity by indexing masks per timestep and pixel.
>
> **Softmax**: As Reviewer 8kUK noted, computing softmax across a dataset is challenging. We address this with a numerically stable streaming implementation requiring just one pass and constant memory (see `src/utils.py` in the appendix files).
>
> **Algorithmic complexity table**:
> |Method|Wiener Filter|Kamb & Ganguli (exact)|Kamb & Ganguli (approx)|Ours (exact, implemented)|Ours (approx, not implemented)|
> |-|-|-|-|-|-|
> |Complexity|$O(m^2)$|$O(n p_t m^2)$|$O\left(\frac{n p_t m^2}{k}\right)$|$O(n p_t m)$| $O\left(\frac{n p_t m}{k}\right)$|
>
>
> ## Choice of $A_t^q$: intuition, other alternatives and formatting
>
> ### Motivation in Section A.3 (8kUK, Ws6m)
>
> We agree with reviewers 8kUK and Ws6m that the main paper will benefit from moving some part of the motivation of our algorithm from the appendix to the main paper. We will update the main paper with a clearer intuition behind our algorithm: Kamb and Ganguli use rectangular binary masks fitted to the sensitivity fields of trained denoisers. In our paper, we show that the sensitivity fields of the trained denoisers come from the data covariance. In this paper, we are considering constant sensitivity fields, i.e., $A_t^q$ is not a function of $x$. If the sensitivity field is constant, the denoiser is linear, and we know that the optimal linear denoiser is the Wiener filter. Thus, the sensitivity field under the constant constraint is that of the Wiener filter.
>
> We also revised the derivations for why the sensitivity fields should be binary:
>
> * **Generalized locality definition** We first consider a generalized notion of locality, with the constraint $f^q(x) = f^q(A^q_t x)$, where $A^q_t$ can be any matrix. Please note that indeed, when $A^q_t$ is invertible, $f^q$ is effectively unconstrained, i.e., there is no locality. When $A^q_t = 0$, $f^q$ does not depend on $x$.
>
> * **Generalized locality imposes subspace projection** Next we prove that imposing the generalized locality constraint is equivalent to adding a projection operator $\mathbf{P_t^q}=(A_t^q)^T (A_t^q (A_t^q)^T)^\dagger A_t^q$ (projection onto the row-space of $A^q_t$) to the optimal denoiser: $\hat{f}^q(x, t) = \sum_{i=1}^N (x^i_0)^q \; \text{softmax}_i\left(-\frac{1}{2\sigma_t^2} \|\mathbf{P_t^q}(x - \sqrt{\alpha_t}x_0^i)\|^2\right)$
>
> * **Projection = Orthogonal change of basis + Masking** All projection operators can be written as an orthogonal change of basis $V_t^q$ followed by a masking operation $M^q_t$ (i.e., a diagonal 0-1 matrix): $\mathbf{P_t^q} = (V_t^q)^TM^q_tV_t^q$
>
> This explains why the only interesting generalized locality matrices are binary masks. We will move motivation for our method and additional intuition from Appendix A.3 to Section 3 of the main paper, in case if the paper is accepted. We will update our paper's appendix with proofs of the above statements.
>
> ### Low-Rank Covariance Matrix
> In response to Reviewer Ws6m’s suggestion, we conducted an additional comparison between the Wiener filter and our algorithm under low-rank projections of the covariance matrix. For the Wiener filter, low-rank approximation leads to visually smoother outputs; however, it reduces the correlation with outputs from trained diffusion models. Our algorithm does not benefit from using a low-rank projection of the covariance matrix and still demonstrates higher correlation than the Wiener filter. This indicates that low-rank structure alone does not account for the performance of learned denoisers. We believe this result further supports the distinctiveness and relevance of our proposed model. Please refer to our detailed qualitative and quantitative analysis in the personal response to Reviewer Ws6m.
>
> # Individual comments
>
> ## RE: the README in the code
> Thank you for pointing this out. We updated our README with detailed installation and execution instructions. We will publicly release a cleaned-up and better-documented codebase together with our camera-ready version.
>
> ## RE: Formatting of Figure 5
> Since the main goal of the work is to study generalization mechanisms of trained diffusion models, we believe it is important to report the nearest image ("OursNN") in the dataset. By showing the closest image, we allow the readers to judge how novel the generated samples are and if the proposed analytical model indeed generalizes. To report the closest images for each baseline while preserving the space in the paper, we plan to change the format of Figure 5 and place a miniature version of the closest neighbor images at the top-right corner of each generated image. We will be happy to follow any recommendations on better formatting of Figure 5. We further agree that the label "OursNN" might be confusing and will rename it to "Dataset" by the camera-ready version.
>
> ## RE: Table 1, moving variance to the main paper
> We will be happy to move the variance for Table 1 from the appendix to the main paper. To comply with NeurIPS formatting, in the camera-ready version of the main paper, we are planning to provide the metrics only on 3 datasets out of 5 (CIFAR10, Celeba-HQ, and MNIST) with the respective variances, while moving the other two datasets (AFHQ and FashionMNIST) to the appendix. Please find the proposed formatting of Table 1 below:
>
> |Method|CIFAR10 r² ↑|CIFAR10 MSE ↓|CelebA-HQ r² ↑|CelebA-HQ MSE ↓|MNIST r² ↑|MNIST MSE ↓|
> |-|-|-|-|-|-|-|
> |Optimal Denoiser|-0.549±0.774|0.191±0.044|0.400±0.298|0.101±0.023|0.187±0.204|0.231±0.036|
> |Wiener (linear)|0.408±0.092|0.032±0.004|0.818±0.039|0.031±0.004|0.469±0.066|0.161±0.014|
> |Kamb & Ganguli (2024)|0.303±0.126|0.065±0.017|0.795±0.081|0.033±0.006|0.283±0.110|0.227±0.045|
> |Niedoba et al. (2024)|0.523±0.137|0.049±0.004|0.733±0.092|0.043±0.005| 0.301±0.077|0.214±0.022|
> |**Ours**|0.589±0.078|0.028±0.008|0.902±0.032|0.016±0.006|0.491±0.051|0.153±0.015|
> |Another DDPM|0.852±0.113|0.023±0.002|0.981±0.007|0.004±0.001|0.969±0.082|0.007±0.019|
>
> ## RE: statistical significance
> Our results are statistically significant. To elucidate this, we calculate the p-value by performing a two-sample t-test against the hypothesis that "Ours" is better than the other baselines. All the results are presented for the $r^2$ coefficient of determination. From the table below, with high statistical significance, our method explains trained diffusion models better than any of the baselines.
>
> |Dataset|vs Wiener|vs Kamb|vs Niedoba|
> |-|-|-|-|
> |CIFAR10|$3.4*10^{-8}$|$8.0*10^{-10}$|$0.0351$|
> |CelebA-HQ|$1.9*10^{-5}$|$6.8*10^{-5}$|$2.9*10^{-9}$|
> |AFHQv2|$5.0*10^{-4}$|$1.1*10^{-6}$|$2.2*10^{-8}$|
> |MNIST|$0.0016$|$8.9*10^{-9}$|$8.0*10^{-4}$|
> |Fashion MNIST|$2.1*10^{-6}$|$9.3*10^{-8}$|$2.6*10^{-8}$|
>
> Please let us know if you have other suggestions on how to format it better.

---

> > ### Comment · Reviewer_4nQU · 2025-08-02
> >
> > I thank the authors for their detailed response and for agreeing to implement the feedback I suggested. I agree that it makes sense to display the nearest neighbour in Figure 5. I think it was just not immediately obvious to me what the purpose of it was. Perhaps a comment (either in the caption or in the main text) that this is to demonstrate that the method is not memorizing training data? I leave this suggestion to your judgment.
> >
> > I will maintain my score of 5 as I do not know the relevant literature well enough to determine if this qualifies as having *ground-breaking impact* to give it a score of 6.

---

> > > ### Author Response · Authors · 2025-08-03
> > >
> > > Thank you for your thorough review and for the score of 5. If there are any specific questions we can help clarify regarding the relevant literature, we'd be happy to do so.

---

### Official Review · Reviewer_8kUK · 2025-07-03

**Clarity:** 3
**Significance:** 2
**Originality:** 3
**Rating:** 5
**Confidence:** 5

**Summary:**

This paper is a nice response to the recent paper Kamb Ganguli, which propose generalization comes from local equivariant inductive bias. This paper showed that the local receptive field is dataset dependent and universal across architectures, suggesting it’s a property of dataset than architecture. They further leveraged the Gaussian linear score solution / Wiener matrix to analytically approximate the receptive field and use it to derive another closed form diffusion sampler.  They showed empirically that their analytical diffusion better approximates actual diffusion model than the Gaussian linear one and local equivariant one.

**Questions:**

### Questions

- Though it’s not the main point of this paper, is there a comparison of efficiency / runtime of your method and Kamb Ganguli method & Wiener filter? In other word Is this new method practical? Wouldn’t the softmax easily run into computational out of memory problems when the dataset goes up?

**Ethical Concerns:**

["NO or VERY MINOR ethics concerns only"]

**Final Justification:**

We strongly recommend this paper as a worthy addition to the literature.
It will clarify some mis-understanding created from previous influential papers and guide development in future works
The new derivation made the method more intuitive, the comparison of computational efficiency, and comparison with Wiener filter with lower rank covariance is interesting and made the work much stronger.

We highly recommend this work for acceptance!

**Limitations:**

N.A.

**Paper Formatting Concerns:**

They forgot to remove the guideline from the checklist.

**Quality:**

3

**Strengths And Weaknesses:**

### Strength

- I agree and am convinced the data statistics place a larger role in determining the sensitivity field. The manipulation of dataset statistics and the UNet vs DiT comparison are very convincing. This is a important and nice point to make given the influential Kamb Ganguli paper.
- Comparison to related works are carefully made.
The qualitative and quantitative benchmark in Table 1 and Figure 5. are nice. Previous work Kamb & Ganguli esp. lack such quantitative and systematic comparison with the Wiener filter results (Wang Vastola 2023, 2024), so it’s good to show quantitatively the linear case works well still.   Using another trained DDPM as reference is also very nice control.

### Weakness

- Maybe in the full paper some of Appendix A and more intuition should be put into main paper to support the Eq. 9? I think the intuition is the using the binarized corresponding qth row of Wiener matrix as the “receptive field” mask.
- Theoretical justification (Supplementary section A.3) it’s a bit confusing to follow.
    - The notation in the  is sometimes not consistent. in line 119 A_t^q is d by d but in the definition in L96 it’s dxd by dxd.
    - The generalized locality definition A.8 does not make much sense for arbitrary linear operators. It cannot be true for a general A_t^q
    - I imagine there is a clearer and more intuitive way to write this section.

---

> ### Author Rebuttal · Authors · 2025-07-31
>
> We thank Reviewer 8kUK for their thoughtful review and for describing our paper as “a nice response” to recent work. We’re glad you found our experimental comparisons “very convincing” and appreciated the “quantitative and systematic comparison with the Wiener filter.” Reviewers 8EDr and Ws6m also found our model to significantly outperform prior analytical approaches, and 4nQU highlighted the clarity and accessibility of our paper.
>
> We also appreciate reviewers' suggestions on how to further strengthen the paper. In the rest of this reply, we first address suggestions shared by multiple reviewers and then will specifically address comments in this review.
>
> ## Computational Efficiency
>
> Reviewers 8EDr and 8kUK suggested including a comparison of runtime between the baselines. The runtimes and hardware used for all baselines are reported in Section C.5 of the appendix, and we provide additional context below. Our algorithm is more computationally efficient than Kamb & Ganguli's model both asymptotically and in practice, as we are not assuming translation equivariance. As reviewer 8kUK correctly mentioned, creating an efficient analytical denoiser is not the main focus of this paper; however, we agree that additional analysis of computational efficiency will further strengthen the paper.
>
> Below, we provide a detailed analysis of the algorithmic complexity for the Wiener filter, Kamb & Ganguli's analytical model, and ours.
>
> ### Algorithmic Complexity and the Softmax
>
> For small-resolution images, the Wiener filter remains the most efficient at $O(m^2)$, where $m$ is the flattened image resolution. Both our model and Kamb & Ganguli’s require a dataset pass per inference step, leading to scaling linear in $n$, where $n$ is the dataset size.
>
> Kamb & Ganguli assume translation equivariance, so for each pixel, we compare its surrounding patch (size $p_t$ at denoising step $t$) to every patch in the dataset, resulting in $O(n p_t m^2)$ complexity. With approximate vector search (e.g., using $k$ clusters), this reduces to $O(n p_t m^2 / k)$. Our model forgoes translation equivariance as we did not observe any difference in quality (reported in the main paper). Additionally, we are using a distinct per-pixel mask pattern. This leads to the algorithmic complexity of our algorithm being $O(n p_t m)$. While this makes approximate search harder, our implementation (Section C.5) is faster than even Kamb & Ganguli’s approximate version on benchmark-scale datasets. For larger datasets, we can match their $O(n p_t m / k)$ complexity by indexing masks per timestep and pixel.
>
> **Softmax**: As Reviewer 8kUK noted, computing softmax across a dataset is challenging. We address this with a numerically stable streaming implementation requiring just one pass and constant memory (see `src/utils.py` in the appendix files).
>
> **Algorithmic complexity table**:
> |Method|Wiener Filter|Kamb & Ganguli (exact)|Kamb & Ganguli (approx)|Ours (exact, implemented)|Ours (approx, not implemented)|
> |-|-|-|-|-|-|
> |Complexity|$O(m^2)$|$O(n p_t m^2)$|$O\left(\frac{n p_t m^2}{k}\right)$|$O(n p_t m)$| $O\left(\frac{n p_t m}{k}\right)$|
>
>
> ## Choice of $A_t^q$: intuition, other alternatives and formatting
>
> ### Motivation in Section A.3 (8kUK, Ws6m)
>
> We agree with reviewers 8kUK and Ws6m that the main paper will benefit from some part of the motivation of our algorithm from the appendix to the main paper. We will update the main paper with a clearer intuition behind our algorithm: Kamb and Ganguli use rectangular binary masks fitted to the sensitivity fields of trained denoisers. In our paper, we show that the sensitivity fields of the trained denoisers come from the data covariance. In this paper, we are considering constant sensitivity fields, i.e., $A_t^q$ is not a function of $x$. If the sensitivity field is constant, the denoiser is linear, and we know that the optimal linear denoiser is the Wiener filter. Thus, the sensitivity field under the constant constraint is that of the Wiener filter.
>
> We also revised the derivations for why the sensitivity fields should be binary:
>
> * **Generalized locality definition** We first consider a generalized notion of locality, with the constraint $f^q(x) = f^q(A^q_t x)$, where $A^q_t$ can be any matrix. Please note that indeed, when $A^q_t$ is invertible, $f^q$ is effectively unconstrained, i.e., there is no locality. When $A^q_t = 0$, $f^q$ does not depend on $x$.
>
> * **Generalized locality imposes subspace projection** Next we prove that imposing the generalized locality constraint is equivalent to adding a projection operator $\mathbf{P_t^q}=(A_t^q)^T (A_t^q (A_t^q)^T)^\dagger A_t^q$ (projection onto the row-space of $A^q_t$) to the optimal denoiser: $\hat{f}^q(x, t) = \sum_{i=1}^N (x^i_0)^q \; \text{softmax}_i\left(-\frac{1}{2\sigma_t^2} \|\mathbf{P_t^q}(x - \sqrt{\alpha_t}x_0^i)\|^2\right)$
>
> * **Projection = Orthogonal change of basis + Masking** All projection operators can be written as an orthogonal change of basis $V_t^q$ followed by a masking operation $M^q_t$ (i.e., a diagonal 0-1 matrix): $\mathbf{P_t^q} = (V_t^q)^TM^q_tV_t^q$
>
> This explains why the only interesting generalized locality matrices are binary masks. We will move motivation for our method and additional intuition from Appendix A.3 to Section 3 of the main paper, in case if the paper is accepted. We will update our paper's appendix with proofs of the above statements.
>
> ### Low-Rank Covariance Matrix
> In response to Reviewer Ws6m’s suggestion, we conducted an additional comparison between the Wiener filter and our algorithm under low-rank projections of the covariance matrix. For the Wiener filter, low-rank approximation leads to visually smoother outputs; however, it reduces the correlation with outputs from trained diffusion models. Our algorithm does not benefit from using a low-rank projection of the covariance matrix and still demonstrates higher correlation than the Wiener filter. This indicates that low-rank structure alone does not account for the performance of learned denoisers. We believe this result further supports the distinctiveness and relevance of our proposed model. Please refer to our detailed qualitative and quantitative analysis in the personal response to Reviewer Ws6m.
>
> Thank you for your thoughtful review, and please let us know if you have any additional suggestions on how we can further improve our paper.

---

> > ### Author Response · Authors · 2025-08-05
> > **Questions Remaining?**
> >
> > Dear reviewer 8kUK,
> > Do you have any remaining questions or concerns following our response? Please let us know, and we would be happy to answer them in the remaining time of the discussion.

---

> ### Comment · Reviewer_8kUK · 2025-08-07
> **Thanks for the rebuttal | additional discussion**
>
> We appreciate the author’s detailed responses and extra work!
>
> **Algorithmic Complexity and the Softmax**
>
> The table comparing the complexities of different methods will be very nice reference for future readers for this line of work. Seems like table like this does not exist in the previous papers, e.g Kamb & Ganguli 2024 and Wang & Vastola 2024).
>
> **Motivation in Section A.3**
>
> Thanks for explaining the Sec. A.3. The generalized locality definition now makes much more sense! (although I still cannot see why this definition is termed “generalized locality” LOL, more like invariance to linear projection / linear transform… )
>
> The intuition of projection equivalent to orthogonal transform + masking makes a lot of sense, and it’s indeed easier to follow.
>
> I think the new logic flow can be briefed in main text to give more intuition!
>
> **Low rank covariance**
>
> Thanks for mentioning the new experiment on low rank covariance. The new experiments and table in response to `Ws6m` is very comprehensive and convincing, Kudos on the effort for pulling this off! :)
>
> The graceful decay of correlation when cutting off spectrum for Wiener filter is kind of expected. Though the degradation of correlation for your new method is less intuitive and sometimes non-monotonic. Is there intuition why that is the case?
>
> From the formula of Wiener filter or in Fig. 10 of [Wang, Vastola, 2024], diffusion at different noise scales should depend on different number PCs in the spectrum, i.e. higher noise scale depend on few number of PCs, and lower noise scales depend nearly all the PCs. So cutting off some lower PCs should only affect the lower noise regimes (at least for Gaussian score / Wiener filter). Not sure if there are some similar intuitions for your new method~
>
> (Not sure keeping the same cutoff rank for all sampling steps will be good. But we also acknowledge that, it will be too much work to tune the rank for different noise levels, so a comprehensive study could be left for future work!)
>
> We'd maintain/increase our supporting score and lean strongly towards acceptance!

---

> > ### Author Response · Authors · 2025-08-09
> > **Thank you for your thoughtful questions and the strong endorsement**
> >
> > ## RE: Algorithmic complexity
> > We will make sure to include the algorithmic complexity analysis in the final version—thank you for the suggestion.
> >
> > ## RE: Section A.3
> > We are happy to hear that our new explanation is easier to follow. We have updated the Appendix and will include a shortened version of the logic flow in the main paper in case of acceptance.
> >
> > ## RE: Term “generalized locality”
> > Let us first elaborate on the formal definition of locality. Identically to Kamb & Ganguli’s work, we call a function $f(x)$ $\Omega$-local if $f^q(M^q_{\Omega} x) = f^q(x)$, where $M^q_{\Omega}$ is a binary mask that selects a region $\Omega$ around pixel $q$ and masks out all other parts of $x$. That is, the function is local if it relies only on a local portion of the input. Note that in this definition:
> > 1. Each pixel $q$ shares the same locality pattern.
> > 2. The masks are binary.
> >
> > Now, in our derivations, we relax both of these constraints and introduce a more flexible matrix $A_t$ in Equation (7) of Definition A.8. Since $A^q_t$ are arbitrary matrices:
> > 1. Each pixel can have a unique locality pattern, as $A^q_t$ can differ across $q$.
> > 2. At the moment we introduce Definition A.8, the locality patterns are not constrained to be binary, as matrices $A^q_t$ are composed of real numbers. Later, we show that relaxing binary masks to real-valued matrices does not change the analytical model; however, at this point, the masks are more general than in the original definition.
> >
> > With this definition, **A.8 still characterizes a function as relying only on a portion of the input, but in a more relaxed and generalized form** than the original; therefore, we decided to call it “generalized locality”.
> >
> > *Please let us know if this intuition makes sense. We are happy to rename the term to make the paper easier to follow.*
> >
> > ## RE: Why does our algorithm’s correlation decay differently from the Wiener filter
> > This is a great observation.
> > We hypothesize that the decay of the correlation of our algorithm is non-monotonic due to the binarization of the covariance matrix.
> > * Wiener filter uses continuous singular values of the covariance matrix, and thus its decay is smooth.
> > * Our algorithm performs binarization to obtain the locality patterns.
> >
> > **Our intuition is that binarization causes non-monotonic, step-like changes in the correlation.** We can include the discussion of this effect in the final paper.
> >
> > ## RE: Adaptive cut-off rank
> > Indeed, we also noticed in our experiments that cutting off lower principal components of the covariance matrix has almost no effect in the high-noise regime ($t > 700$) and mostly plays a role in the low-noise regime (for both the Wiener filter and our algorithm).
> >
> > A quick experiment tuning the cut-off ranks for each noise level did not show a significant improvement. We agree, however, that it makes sense for the cut-off ranks to vary across different noise levels. We believe that a more rigorous analysis is required to reach conclusive results, and we will leave this to future work. Thank you for the suggestion.
> >
> > ---
> > Thank you for your encouragement and detailed review! Even though the discussion period is coming to an end, please feel free to recommend any additional improvements to the clarity of the paper.

---

### Note · Authors · 2025-08-15

Dear reviewers and AC,

Thank you for the time and effort spent reviewing our work.

We are grateful for the unanimously positive evaluation of our paper. Reviewer 8EDr noted our approach "surprisingly outperforms previous analytical models" with "new quantitative benchmarks" and "detailed visual proofs." Reviewer Ws6m appreciated that our findings "correct the common belief" [that the inductive bias is a reason for generalization in image diffusion models] and improve upon previous models, "contributing to the understanding of generalization mechanisms." Reviewer 8kUK called the paper "a nice response" to recent work, highlighting our "very convincing" experimental argumentation. Reviewer 4nQU described the paper as "very well-written" with "clearly explained," "convincing" experiments and "no major weaknesses."

We also thank the reviewers for the constructive feedback that we addressed in the rebuttal:
1. *Computational Complexity*: Provided a detailed algorithmic complexity comparison showing our method is more efficient than [Kamb & Ganguli 2024] as we dont require shift-equivariance
2. *Extended Theoretical Motivation*: Committed to moving key intuitions from the appendix to the main paper for better accessibility, and to rewriting the derivation of the binary mask in the analytical model
3. *Low-Rank Projection*: Conducted new experiments studying how projecting the data covariance matrix to a lower rank affects correlation with the scores of a trained U-Net
4. *Statistical Significance*: Confirmed all results are statistically significant (p < 0.05)
5. *Improved Formatting*: Addressed visualization and table formatting suggestions

Our work provides important insights into the inner workings of generative diffusion models, providing theoretical insights into the mechanisms of generating new samples; the findings correct previous beliefs and allow us to propose an augmented algorithm with better performance. Furthermore, we introduce extensive comparisons with such baselines as the Wiener filter that did not appear in previous work, which are crucial for understanding the predictive power of analytical models.

Best regards,

Authors

---

### Decision · Program_Chairs · 2025-09-17

**Decision:**

Accept (spotlight)

**Comment:**

This paper received three accept ratings and one borderline accept rating. The reviewers found several weaknesses, which were addressed by the rebuttal comments. The paper is well-written, addresses an important topic and the experiments are convincing, which justify an accept decision on behalf of the AC.